# EFFICIENT SCORE MATCHING WITH DEEP EQUILIBRIUM LAYERS

**Yuhao Huang**[1], **Qingsong Wang**[1], **Akwum Onwunta**[2], **& Bao Wang**[1]*
[1]Department of Mathematics and Scientific Computing and Imaging (SCI) Institute
University of Utah, Salt Lake City, UT 84102, USA
[2]Department of Industrial and Systems Engineering
Lehigh University, Bethlehem, PA 18015, USA

## ABSTRACT

Score matching methods, which estimate probability densities without computing the normalization constant, are particularly useful in deep learning. However, the computational and memory costs of score matching methods can be prohibitive for high-dimensional data or complex models, particularly due to the derivatives or Hessians of the log density function appearing in the objective function. Some existing approaches modify the objective function to reduce the quadratic computational complexity for the Hessian computation. However, the memory bottleneck of score matching methods remains for deep learning. This study improves the memory efficiency of score matching by leveraging deep equilibrium models. We provide a theoretical analysis of deep equilibrium models for scoring matching and applying implicit differentiation to higher-order derivatives. Empirical evaluations demonstrate that our approach enables the development of deep and expressive models with improved performance and comparable computational and memory costs over shallow architectures.

## 1 INTRODUCTION

Score matching [16] and its variants [33; 34; 43; 40] is a class of density estimation methods, which avoid computing the normalization constant of the density function; see Section 2.1 for a brief review of score matching methods. Score matching is particularly useful when the density function is parameterized by a deep neural network (DNN). At its core, score matching utilizes the fact that the (Stein) score function – the gradient of log density function – does not require the computation of the normalization constant; one can estimate the density function by minimizing the discrepancy between the score functions of the true data distribution and the model distribution. Given its flexibility, score matching methods have been applied to various tasks requiring density estimation [19; 43; 44; 37] – especially diffusion models; see e.g., [37; 38; 14; 41].

Despite the remarkable advantages of score matching, its applications to DNNs face computational and memory challenges, arising from optimizing the objective function of the score matching method, which necessitates the computation of the derivative and Hessian of the log density function [16; 39]. This step, coupled with the subsequent backpropagation, which additionally computes the gradient with respect to the weights, significantly inflates the computational graph for training DNNs [19; 25; 39; 40].

Several strategies have been proposed to address the computational and memory challenges of score matching methods, focusing on reducing the quadratic computational cost of the Hessian. For instance, methods such as approximate backpropagation [19] and curvature propagation (CP) [25] provide estimations for the Hessian. Additionally, there are efforts to modify the score matching objective functions, including denoising score matching (DSM) [43], which avoids the Hessian computation by considering an objective function with a perturbed data distribution; see equation 2. Likewise, the sliced score matching (SSM) [40] is proposed to replace the Hessian matrix computation with the Hessian vector product; see equation 3. These methods alleviate the computational cost of the Hessian term in the score matching methods, but the memory bottleneck that arises from optimizing the objective function that involves derivatives of the log density function remains. This

---

*Correspond to `wangbaonj@gmail.com`

memory bottleneck restricts the depth and complexity of models, limiting their potential in deep learning applications. For example, in Section 4.1, a VAE variant is trained with SSM objective function; while increased model depth offers noticeable performance gains, it comes at a substantial cost of memory and computational overhead; see Table 1 for details. Deep equilibrium model (DEQ) [3; 45] presents an attractive solution to address these challenges. In DEQ, the hidden layer is represented as the equilibrium point of a nonlinear fixed-point equation that is equivalent to a weight-tied infinite-depth neural network [45]. Importantly, by applying the implicit function theorem [1; 8] at this equilibrium, DEQ facilitates the computation of the derivatives of the hidden representation without referencing intermediate activation values. As a result, DEQ enjoys constant memory efficiency while being a deep and expressive model [15].

Inspired by the memory efficiency and remarkable performance of DEQ [3; 45; 4; 7; 31; 2; 32; 27], we propose to address the computational and memory bottlenecks of score matching methods by leveraging DEQ. Using the implicit function theorem, the score function can be computed in terms of the fixed point of the DEQ without storing the intermediate activation values. However, this approach requires computing the inverse of the Jacobian matrix of the DEQ, which can be expensive. We then use an efficient implementation of the backpropagation that utilizes the so-called phantom gradient [12] in the presence of higher-order derivatives. As a result, the proposed DEQ-based score matching methods allow for the design of deep and expressive models with computational and memory costs comparable to those of shallow architectures. We demonstrate the effectiveness of the proposed method through empirical evaluations of both density estimation and generative modeling.

## 1.1 OUR CONTRIBUTIONS
We present a systematic study on integrating DEQ into density estimation tasks or when the loss function involves derivatives in general. Our work makes the following key contributions:

- We integrate DEQs into score matching models to address their memory challenges.
- We provide an efficient implementation of the backpropagation and analyze its convergence in the presence of higher-order derivatives.
- Through empirical studies, we confirm that the integration of DEQs allows for the design of deeper and more expressive models without elevating computational and memory demands

## 1.2 ADDITIONAL RELATED WORKS
Beyond the score function, some works consider higher-order derivatives of the log density function [29]. For example, [26] terms these derivatives as higher-order scores and presents a generalization of DSM. FDSSM, introduced in [28], adopts finite difference to approximate the directional derivatives of the log density function. It uses random projection to approximate the gradient terms. As observed in their experiments [28, Section 6.1], the variance introduced by the random projection can outweigh the computational benefits of the finite difference approximation.

Another additional line of related work is using DEQ to improve the memory efficiency of deep learning models. [15] combines DEQ with the implicit representations to achieve improved performance with less memory. In optical flow estimations, DEQ model [7] demonstrates faster computation of the flow, significantly reduced memory needs, and improves over state-of-the-art models.

## 1.3 ORGANIZATION

This paper is organized as follows: Section 2 provides an overview of the score matching method and its challenges, followed by an introduction to DEQ. In Section 3, we present our proposed method for integrating DEQs into score matching models. Specifically, Section 3.1 describes DEQ for score matching models, Section 3.2 discusses the well-posedness of the fixed-point equation, and Section 3.3 presents the implicit differentiation for higher-order derivatives. Section 3.4 describes the efficient implementation of backpropagation in the presence of higher-order derivatives. Section 4 presents the empirical evaluations of the proposed method on density estimation and generative modeling tasks.

## 2 BACKGROUND AND PRELIMINARIES

This section provides an overview of the score matching method and its challenges, followed by an introduction to DEQ.

### 2.1 SCORE MATCHING FOR UNNORMALIZED DENSITY ESTIMATION

In machine learning, statistical models often rely on unnormalized density function $\tilde{p}_{\boldsymbol{\theta}}(\cdot)$, which is proportional to the normalized (probability) density function $p_{\boldsymbol{\theta}}(\cdot) = \frac{1}{Z_\theta}\tilde{p}_{\boldsymbol{\theta}}(\cdot)$, where $Z_\theta$ is the nor-

malization constant. The normalization constant depends on the parameter $\boldsymbol{\theta}$ and is often intractable to compute. Thus the maximum likelihood estimation, which minimizes the Kullback-Leibler (KL) divergence between the true data density $p_{\text{data}}(\cdot)$ and the model density $\tilde{p}_{\boldsymbol{\theta}}(\cdot)$ is infeasible.

**Score matching (SM).** Instead of using KL divergence to quantify the discrepancy between the true density $p_{\text{data}}(\cdot)$ and the normalized density $p_{\boldsymbol{\theta}}(\cdot)$, one can alternatively use the Fisher divergence $\mathcal{D}_F$ that compares the score functions $\nabla_{\boldsymbol{x}} \log p_{\text{data}}(\boldsymbol{x})$ and $\nabla_{\boldsymbol{x}} \log \tilde{p}_{\boldsymbol{\theta}}(\boldsymbol{x})$ of the two densities:

$$
\begin{aligned}
\mathcal{D}_F(p_{\boldsymbol{\theta}}, p_{\text{data}}) &:= \frac{1}{2} \mathbb{E}_{\boldsymbol{x} \sim p_{\text{data}}} \left[ \| \nabla_{\boldsymbol{x}} \log p_{\boldsymbol{\theta}}(\boldsymbol{x}) - \nabla_{\boldsymbol{x}} \log p_{\text{data}}(\boldsymbol{x}) \|^2 \right] \\
&= \frac{1}{2} \mathbb{E}_{\boldsymbol{x} \sim p_{\text{data}}} \left[ \| \nabla_{\boldsymbol{x}} \log \tilde{p}_{\boldsymbol{\theta}}(\boldsymbol{x}) - \nabla_{\boldsymbol{x}} \log p_{\text{data}}(\boldsymbol{x}) \|^2 \right] \\
&= \mathcal{D}_F(\tilde{p}_{\boldsymbol{\theta}}, p_{\text{data}}),
\end{aligned}
$$

showing that Fisher divergence avoids the computation of the normalization constant $Z_\theta$. However, it requires computing the score function of the true data density $p_{\text{data}}(\cdot)$, which is inaccessible.

Based on some regularity assumption of the data density $p_{\text{data}}(\cdot)$, Hyvärinen [16] finds that the Fisher divergence can be rewritten as $\mathcal{D}_F = \mathcal{J}_{\text{SM}} + C$, where $C$ is a constant independent of $\boldsymbol{\theta}$ and $\mathcal{J}$ is defined as follows:

$$
\mathcal{J}_{\text{SM}}(\boldsymbol{\theta}) := \frac{1}{2} \mathbb{E}_{\boldsymbol{x} \sim p_{\text{data}}} \left[ \| \nabla_{\boldsymbol{x}} \log \tilde{p}_{\boldsymbol{\theta}}(\boldsymbol{x}) \|^2 \right] + \mathbb{E}_{\boldsymbol{x} \sim p_{\text{data}}} \left[ \text{tr} \left( \nabla_{\boldsymbol{x}}^2 \log \tilde{p}_{\boldsymbol{\theta}}(\boldsymbol{x}) \right) \right], \tag{1}
$$

where $\text{tr}(\cdot)$ denotes the trace operator. Then, the proposed score matching method in [16] finds the optimal parameters $\hat{\boldsymbol{\theta}}$ by minimizing the objective function $\mathcal{J}_{\text{SM}}$.

Despite its theoretical advantages, the score matching method faces computational and memory challenges when using DNNs to model the score function. The challenge comes from the computation of the derivative and the trace of Hessian terms in the objective function $\mathcal{J}_{\text{SM}}$. For example, the common implementation of automatic differentiation like PyTorch [30] requires computing the full Hessian matrix $\nabla_{\boldsymbol{x}}^2 \log \tilde{p}_{\boldsymbol{\theta}}(\boldsymbol{x})$ even when we only need its trace. The quadratic computational complexity of the Hessian matrix computation is prohibitive for high-dimensional data or deep models. Furthermore, when optimizing the objective function $\mathcal{J}_{\text{SM}}$ with backpropagation, it requires taking the derivative of the score function $\nabla_{\boldsymbol{x}} \log \tilde{p}_{\boldsymbol{\theta}}(\boldsymbol{x})$ and the Hessian matrix $\nabla_{\boldsymbol{x}}^2 \log \tilde{p}_{\boldsymbol{\theta}}(\boldsymbol{x})$, which necessitates storing their computational graph which leads to prohibitive memory costs. Several methods have been proposed to reduce the computational and memory costs of score matching.

**Denoising Score Matching (DSM).** [43] proposes to perturb the data $\boldsymbol{x}$ according to a noise distribution $p_\sigma(\tilde{\boldsymbol{x}} | \boldsymbol{x})$ and then estimate the score function of the perturbed data $\tilde{\boldsymbol{x}}$. Specifically, when the noise distribution is Gaussian with variance $\sigma^2$, DSM minimizes the following objective function:

$$
\mathcal{J}_{\text{DSM}}(\boldsymbol{\theta}) := \frac{1}{2} \mathbb{E}_{\boldsymbol{x} \sim p_{\text{data}}} \mathbb{E}_{p_\sigma(\tilde{\boldsymbol{x}} | \boldsymbol{x})} \left[ \left\| \nabla_{\tilde{\boldsymbol{x}}} \log \tilde{p}_{\boldsymbol{\theta}}(\tilde{\boldsymbol{x}}) + \frac{\tilde{\boldsymbol{x}} - \boldsymbol{x}}{\sigma^2} \right\|^2 \right]. \tag{2}
$$

This method avoids the computation of the Hessian matrix $\nabla_{\boldsymbol{x}}^2 \log \tilde{p}_{\boldsymbol{\theta}}(\boldsymbol{x})$ but at the cost of recovering the perturbed data $p_\sigma(\tilde{\boldsymbol{x}})$ instead of the true data distribution $p_{\text{data}}(\boldsymbol{x})$. Meanwhile, the model is sensitive to the parameter $\sigma$ and requires heuristics and careful selection of the value of $\sigma$ [40; 35].

**Sliced Score Matching (SSM).** [40] proposes a random projection-based approach for efficient score matching by replacing computing the Hessian matrix $\nabla_{\boldsymbol{x}}^2 \log \tilde{p}_{\boldsymbol{\theta}}(\boldsymbol{x})$ with Hessian vector products. SSM minimizes the following objective function:

$$
\mathcal{J}_{\text{SSM}}(\boldsymbol{\theta}) := \mathbb{E}_{\boldsymbol{v}} \mathbb{E}_{\boldsymbol{x} \sim p_{\text{data}}} \left[ \frac{1}{2} \left( \boldsymbol{v}^\top \nabla_{\boldsymbol{x}} \log \tilde{p}_{\boldsymbol{\theta}}(\boldsymbol{x}) \right)^2 + \boldsymbol{v}^\top \nabla_{\boldsymbol{x}}^2 \log \tilde{p}_{\boldsymbol{\theta}}(\boldsymbol{x}) \boldsymbol{v} \right], \tag{3}
$$

where $\boldsymbol{v}$ is a random vector sampled from the normal distribution or Rademacher distribution. The term $\mathbb{E}_{\boldsymbol{v}} \frac{1}{2} \left( \boldsymbol{v}^\top \nabla_{\boldsymbol{x}} \log \tilde{p}_{\boldsymbol{\theta}}(\boldsymbol{x}) \right)^2$ can be computed analytically, and it equals to $\frac{1}{2} \| \nabla_{\boldsymbol{x}} \log \tilde{p}_{\boldsymbol{\theta}}(\boldsymbol{x}) \|_2^2$. This leads to SSM with the variance reduction (SSM-VR) objective function [40]:

$$
\mathcal{J}_{\text{SSM-VR}}(\boldsymbol{\theta}) := \mathbb{E}_{\boldsymbol{v}} \mathbb{E}_{\boldsymbol{x} \sim p_{\text{data}}} \left[ \boldsymbol{v}^\top \nabla_{\boldsymbol{x}}^2 \log \tilde{p}_{\boldsymbol{\theta}}(\boldsymbol{x}) \boldsymbol{v} \right] + \frac{1}{2} \| \nabla_{\boldsymbol{x}} \log \tilde{p}_{\boldsymbol{\theta}}(\boldsymbol{x}) \|_2^2. \tag{4}
$$

Even though the above methods reduce the computational cost of the Hessian matrix, they still include derivative in terms with respect to the data $\boldsymbol{x}$ in the objective function, which requires storing the computational graph for backpropagation and leads to computational and memory bottlenecks.

## 2.2 DEEP EQUILIBRIUM NETWORK

DEQ, first presented in [3], represents the hidden representation, $\boldsymbol{z}^*$, as the equilibrium of a specific fixed-point iteration equation:

$$\boldsymbol{z}^* = f_{\boldsymbol{\theta}}(\boldsymbol{z}^*, \boldsymbol{x}),$$

where $f_{\boldsymbol{\theta}}$ represents a neural network parameterized by $\boldsymbol{\theta}$, and $\boldsymbol{x}$ denotes the input data. When using an iterative solver for this fixed-point equation, the equilibrium of the DEQ corresponds to the final hidden representation of an infinite-depth neural network. By leveraging implicit differentiation during training, DEQ eliminates the need to store intermediate hidden representations; results in constant memory usage despite being a deep model – a significant computational advantage. The theoretical convergence of DEQ has been extensively studied in [45]. Further applications prove the model's versatility and competitive performance across various tasks. These include language modeling [3], semantic segmentation [4], optical flow [7], diffusion models [31], and maximum a-posteriori (MAP) estimates [42].

## 3 DEEP EQUILIBRIUM LAYERS FOR SCORE MATCHING

In existing score-based density estimation using DNNs, the output of the DNN is typically the log density function, and the Fisher divergence-related objective functions are optimized. These objective functions often involve the derivatives of the log density function and necessitate the computation of higher-order derivatives in backpropagation. We propose to use DEQ for the core intermediate layer of the DNN architecture that accounts for the main depth and complexity of the model. Then, we can utilize the memory efficiency of DEQ to reduce the overall memory consumption.

### 3.1 GENERAL STRUCTURE OF DEEP EQUILIBRIUM LAYERS

The general structure of our DEQ-assisted score matching model is outlined as follows: An input data, $\boldsymbol{x}$, first being transformed through a simple multi-layer perceptron (MLP). Subsequently, it is processed by a deep equilibrium layer, which finds the fixed point $\boldsymbol{z}^*$ of the following equation:

$$\boldsymbol{z}^{(n+1)} = f_{\boldsymbol{\theta}}(\boldsymbol{z}^{(n)}, \boldsymbol{x}),$$

where $f_{\boldsymbol{\theta}}$ is a neural network parameterized by $\boldsymbol{\theta}$. The computed fixed point, $\boldsymbol{z}^*$, is then converted through a simple network into the log density, $\log \tilde{p}_{\boldsymbol{\theta}}(\boldsymbol{z}^*)$ or score function, $\nabla \log \tilde{p}_{\boldsymbol{\theta}}(\boldsymbol{z}^*)$, depending on the task. Therefore, the main computational demands revolve around DEQ.

Specifically, the iterative functions $f_{\boldsymbol{\theta}}$ consist of one or few sequential blocks of the following form:

$$\boldsymbol{z}^{(n+1)} = \sigma(\boldsymbol{W}\boldsymbol{z}^{(n)} + g(\boldsymbol{y})), \tag{5}$$

with $\sigma$ being an element-wise activation function, $g(\boldsymbol{y})$ is the transformed input for this block, and $\boldsymbol{W}$ is the weight matrix. This structure can represent either a fully connected layer when $\boldsymbol{W}$ is a dense matrix or a convolutional layer when $\boldsymbol{W}$ is a convolutional kernel. Therefore, it provides a flexible framework for adapting the existing density or score estimate models into DEQ.

### 3.2 THE WELL-POSEDNESS OF THE FIXED-POINT EQUATION

In order to correctly apply DEQ, the fixed-point equation must be well-posed; that is, the fixed point $\boldsymbol{z}^*$ must exist and be unique. The well-posedness of the fixed-point equation 5 can be guaranteed by constraining the Frobenius norm of the weight matrix $\boldsymbol{W}$ to be less than one. In particular, we apply the following weight normalization to $\boldsymbol{W}$: $\boldsymbol{W} \rightarrow \boldsymbol{W}/\lambda(\|\boldsymbol{W}\|_{\mathrm{F}} + \epsilon)$, where $\lambda > 1$ is hyperparameter, $\epsilon$ is a small positive number, and $\|\boldsymbol{W}\|_{\mathrm{F}}$ denotes the Frobenius norm of $\boldsymbol{W}$. The computational overhead of this normalization is minimal. This constraint ensures the spectral norm of the weight matrix $\boldsymbol{W}$ to be less than one, and hence the map $\boldsymbol{z} \rightarrow \boldsymbol{W}\boldsymbol{z} + g(\boldsymbol{y})$ is a contraction. Given most activation functions, such as $\mathrm{Softplus}$ used in equation 5 are non-expansive, the composite map $\boldsymbol{z} \rightarrow \sigma(\boldsymbol{W}\boldsymbol{z} + g(\boldsymbol{y}))$ is also a contraction mapping in $\boldsymbol{z}$ and so is $f_{\boldsymbol{\theta}}(\boldsymbol{z}, \boldsymbol{x})$. Thus, Banach's fixed-point theorem [46] guarantees the fixed point's well-posedness. Furthermore, the direct Picard iteration will converge linearly to the fixed point $\boldsymbol{z}^*$.

### 3.3 IMPLICIT DIFFERENTIATION FOR HIGHER-ORDER DERIVATIVES

In this subsection, we describe the computation of the score or its derivatives with respect to the input data $\boldsymbol{x}$ in the presence of the DEQ component. Let $\boldsymbol{x} = (x_1, x_2, \cdots, x_d)^\top \in \mathbb{R}^d$ be the input data. We use $\boldsymbol{z}^* \in \mathbb{R}^n$ to denote the fixed point of DEQ. Then the $i$-th component of the score function $\nabla_{\boldsymbol{x}} \log \tilde{p}_{\boldsymbol{\theta}}(\boldsymbol{z}^*)$ for $1 \leq i \leq d$ can be computed as follows:

$$\frac{\partial \log \tilde{p}_{\boldsymbol{\theta}}(\boldsymbol{z}^*)}{\partial x_i} = \frac{\partial \log \tilde{p}_{\boldsymbol{\theta}}(\boldsymbol{z}^*)}{\partial \boldsymbol{z}^*} \frac{\partial \boldsymbol{z}^*}{\partial x_i}, \tag{6}$$

Likewise, the second-order derivative of the log density function $\frac{\partial^2 \log \tilde{p}_{\boldsymbol{\theta}}(\boldsymbol{z}^*)}{\partial x_i \partial x_j}$ for $1 \leq i, j \leq d$ can be computed as follows:

$$\frac{\partial^2 \log \tilde{p}_{\boldsymbol{\theta}}(\boldsymbol{z}^*)}{\partial x_i \partial x_j} = \frac{\partial^2 \log \tilde{p}_{\boldsymbol{\theta}}(\boldsymbol{z}^*)}{\partial \boldsymbol{z}^* \partial \boldsymbol{z}^*} \frac{\partial \boldsymbol{z}^*}{\partial x_i} \frac{\partial \boldsymbol{z}^*}{\partial x_j} + \frac{\partial \log \tilde{p}_{\boldsymbol{\theta}}(\boldsymbol{z}^*)}{\partial \boldsymbol{z}^*} \frac{\partial^2 \boldsymbol{z}^*}{\partial x_i \partial x_j}. \tag{7}$$

Therefore, the main computational challenge is to compute the term $\frac{\partial \boldsymbol{z}^*}{\partial x_i}$ and $\frac{\partial^2 \boldsymbol{z}^*}{\partial x_i \partial x_j}$. For this, we apply the implicit function theorem to obtain the following result:

**Proposition 1.** *Given $f_{\boldsymbol{\theta}}(\boldsymbol{z}, \boldsymbol{x})$, a continuously differentiable function that is a contraction mapping. Let $\boldsymbol{z}^*$ is the fixed point of the equation $f_{\boldsymbol{\theta}}(\boldsymbol{z}^*, \boldsymbol{x}) = \boldsymbol{z}^*$, then the matrix $\boldsymbol{I} - \left.\frac{\partial f_{\boldsymbol{\theta}}}{\partial \boldsymbol{z}}\right|_{\boldsymbol{z}^*}$ is invertible. Furthermore, the derivative of $\boldsymbol{z}^*$ with respect to $\boldsymbol{x}$ can be computed as follows:*

$$\frac{\partial \boldsymbol{z}^*}{\partial x_i} = \left( \boldsymbol{I} - \left.\frac{\partial f_{\boldsymbol{\theta}}}{\partial \boldsymbol{z}}\right|_{\boldsymbol{z}^*} \right)^{-1} \left.\frac{\partial f_{\boldsymbol{\theta}}}{\partial x_i}\right|_{\boldsymbol{z}^*}, \tag{8}$$

*and if $f_{\boldsymbol{\theta}}$ is additionally twice continuously differentiable, the second-order derivative of $\boldsymbol{z}^*$ with respect to $\boldsymbol{x}$ can be computed as follows:*

$$\frac{\partial^2 \boldsymbol{z}^*}{\partial x_i \partial x_j} = \left( \boldsymbol{I} - \left.\frac{\partial f_{\boldsymbol{\theta}}}{\partial \boldsymbol{z}}\right|_{\boldsymbol{z}^*} \right)^{-1} \left( \left.\frac{\partial^2 f_{\boldsymbol{\theta}}}{\partial x_i \partial x_j}\right|_{\boldsymbol{z}^*} + \left.\frac{\partial^2 f_{\boldsymbol{\theta}}}{\partial \boldsymbol{z} \partial x_j}\right|_{\boldsymbol{z}^*} \frac{\partial \boldsymbol{z}^*}{\partial x_i} + \left.\frac{\partial^2 f_{\boldsymbol{\theta}}}{\partial \boldsymbol{z} \partial x_i}\right|_{\boldsymbol{z}^*} \frac{\partial \boldsymbol{z}^*}{\partial x_j} + \left.\frac{\partial^2 f_{\boldsymbol{\theta}}}{\partial \boldsymbol{z} \partial \boldsymbol{z}}\right|_{\boldsymbol{z}^*} \frac{\partial \boldsymbol{z}^*}{\partial x_i} \frac{\partial \boldsymbol{z}^*}{\partial x_j} \right). \tag{9}$$

To the best of our knowledge, equation 8 is well-known in the DEQ literature; see e.g. [3], while equation 9 is new in DEQ context. In practice, the Softplus activation function is used in DNNs to approximate the log density function, ensuring $f_{\boldsymbol{\theta}}$ is twice continuously differentiable [40]. When using the reverse-mode automatic differentiation, the score function and its derivatives can be computed by only using the states of the DEQ at the fixed point $\boldsymbol{z}^*$.

## 3.4 EFFICIENT IMPLEMENTATION WITH DEQ

There have been efforts devoted to improving training DEQs, including the use of Anderson acceleration [3] and a separate neural block [5] to accelerate the fixed-point finding. Effective techniques have been employed for the forward pass, such as reusing the fixed point from previous iterations [7; 15], Jacobian regularization [6]. Meanwhile, the inexact gradient [12] can render the computational cost of backpropagation negligible without sacrificing the model's performance [7; 2; 31]. In our DEQs, we adopt the strategy of reusing the fixed point from prior iterations to accelerate the forward pass. Additionally, we use inexact gradients, which we will describe in detail in this subsection. Particularly, we will analyze its behavior for higher-order derivatives.

Proposition 1 highlights DEQ's capability in computing the score function and its derivatives without storing the intermediate activation values. Yet, the computation of the matrix inversion $\left( \boldsymbol{I} - \left.\frac{\partial f_{\boldsymbol{\theta}}}{\partial \boldsymbol{z}}\right|_{\boldsymbol{z}^*} \right)^{-1}$ can be expensive. Inspired by the success of inexact gradient in training DEQs, we adopt the phantom gradient [12; 2; 31] for computing the first and second-order derivatives of the score function. The implementation of phantom gradient takes the following two steps:

Step 1 Compute the fixed point $\boldsymbol{z}^*$ of the DEQ by performing forward iterations until convergence with automatic differentiation turned off.

Step 2 Perform $K$ additional forward iterations from the fixed point $\boldsymbol{z}^*$ with automatic differentiation turned on.

We provide a PyTorch-style pseudocode for this implementation in Algorithm B in the appendix. Since we need to perform multiple forward iterations to compute the fixed point $\boldsymbol{z}^*$, the additional forward iterations in Step 2 only incur a small computational overhead. Proposition 2 shows that the derivative with respect to the input data computed by this method converges to the true derivatives as the number of forward iterations $K$ increases. In our experiments, we set $K = 2$ and find that the models already achieve good performance, and this aligns with findings in training DEQs [31; 2; 7].

**Proposition 2.** *Consider function $f_{\boldsymbol{\theta}}(\boldsymbol{z}, \boldsymbol{x})$ with fixed point $\boldsymbol{z}^*$. Let $\{\boldsymbol{z}^{(k)} | 1 \leq k \leq K\}$ be generated by performing $K$ forward iterations from $\boldsymbol{z}^*$, i.e., set $\boldsymbol{z}^{(0)} = \boldsymbol{z}^*$ and $\boldsymbol{z}^{(k+1)} = f_{\boldsymbol{\theta}}(\boldsymbol{z}^{(k)}, \boldsymbol{x})$ for $1 \leq k \leq K$. Let $\boldsymbol{J} = \left.\frac{\partial f_{\boldsymbol{\theta}}}{\partial \boldsymbol{z}}\right|_{\boldsymbol{z}^*}$. Then the derivative $\left.\frac{\partial \boldsymbol{z}^{(K)}}{\partial x_i}\right|_{\boldsymbol{z}^*}$ matches approximation of derivative of the implicit function $\boldsymbol{z}^*$ in $x_i$ by using $K$-th order Neumann series of $(\boldsymbol{I} - \boldsymbol{J})^{-1}$, i.e.,*

$$\frac{\partial \boldsymbol{z}^{(K)}}{\partial x_i} = \sum_{k=0}^{K-1} \boldsymbol{J}^k \left.\frac{\partial f_{\boldsymbol{\theta}}}{\partial x_i}\right|_{\boldsymbol{z}^*}. \tag{10}$$

*Hence as $K \to \infty$, the derivative $\frac{\partial z^{(K)}}{\partial x_i}\Big|_{z^*}$ converges to the true derivative, i.e.,*

$$\lim_{K \to \infty} \frac{\partial z^{(K)}}{\partial x_i} = \frac{\partial z^{(*)}}{\partial x_i}.$$

*Similarly, if $f_\theta$ is twice continuously differentiable, the second-order derivative $\frac{\partial^2 z^{(K)}}{\partial x_i \partial x_j}$ at $z^*$ satisfies*

$$\lim_{K \to \infty} \frac{\partial^2 z^{(K)}}{\partial x_i \partial x_j} = \frac{\partial^2 z^{(*)}}{\partial x_i \partial x_j}.$$

Again, the regularity assumption in the above propositions aligns with the assumption in score matching based models [16; 40]. Moreover, as far as we are aware, the second-order results in the above proposition are new in the DEQ context.

## 4 EXPERIMENTS

In this section, we evaluate the performance of the proposed DEQ-assisted score matching models, including score matching variational autoencoder (SMVAE) [40] in Section 4.1 and noise conditional score network (NCSN) [40] in Section 4.4 for generative modeling. We also consider deep kernel exponential families (DKEF) [44] in Section 4.2 and nonlinear independent components estimation (NICE) [10] in Section 4.3 for density estimation. In our experiments, we utilize the open source code from [40; 37] and adopt the same training setup for fair comparisons.

### 4.1 SCORE ESTIMATION IN VAE WITH IMPLICIT DISTRIBUTION

VAE [18] learns a latent variable $z$ from the observed data $x$, which contains a decoder $p_\theta(x|z)$ that models the conditional distribution of $x$ given the latent variable $z$ and an encoder $q_\phi(z|x)$ that approximates the posterior distribution of the latent variable $z$. A VAE model is trained by maximizing the following evidence lower bound (ELBO):

$$\mathcal{L}(\theta, \phi) = \mathbb{E}_{x \sim p_{\text{data}}} \left[ \mathbb{E}_{z \sim q_\phi(z|x)} [\log p_\theta(x|z)] - \mathbb{E}_{z \sim q_\phi(z|x)} [\log q_\phi(z|x)] \right]. \tag{11}$$

Therefore, a typical training procedure assumes that $q_\phi(z|x)$ is a simple distribution to make the computation tractable. Following [40, Section 6.2.1], the score estimation techniques can be utilized to compute the term $\nabla_\phi \mathbb{E}_{z \sim q_\phi(z|x)} [\log q_\phi(z|x)]$ directly thereby allowing $q_\phi(z|x)$ to be an *implicit distribution* – a distribution without tractable density. In [40], a variant of VAE is proposed that employs a score estimator network to facilitate the ELBO computation, and the score estimator network is trained by minimizing the SSM objective function $\mathcal{J}_{\text{SSM}}$ in equation 3; we refer to this model as SSM VAE; see Table 6 in the appendix for the detailed model architecture.

We consider two image generation tasks: CelebA [24] and Cifar10 [20]. CelebA is a dataset that contains $64 \times 64 \times 3$ color images that identify celebrity face attributes, and Cifar10 contains 10 classes of $32 \times 32 \times 3$ color images. Each dataset is split into train, validation, and test sets with 70%, 20%, and 10%, respectively. In our experiments, we examine the performance of SSM VAE regarding its score estimator network depth. We augment the score estimator network with additional $k$ convolutional layers and term the model as Aug($k$)-SSM VAE, $k = 8$ or 16. We also turn the augmented CNN layer into a (convolutional) DEQ block and term the model as DEQ-SSM VAE. The detailed model architecture is shown in Table 7 in Appendix D.3. We train the models using Adam [17], for $10^5$ iterations, with learning rate $1e$-4, weight decay $1e$-12, and batch size 128.

Figure 1 and Table 1 compare the performance of SSM VAE, Aug($k$)-SSM VAE, and DEQ-SSM VAE in terms of SSM loss, ELBO, FID score [13], and memory usage. The comparisons between SSM VAE and Aug($k$)-SSM VAE show that increasing the number of encoding layers improves the model's performance, but it also significantly increases the memory usage and training time. In contrast, DEQ-SSM VAE outperforms the other models in terms of SSM loss, ELBO, and FID score while costing the smallest amount of memory overhead compared to the baseline SSM VAE model.

### 4.2 DEEP KERNEL EXPONENTIAL FAMILIES FOR DENSITY ESTIMATION

DKEF is an unnormalized density estimation model proposed in [44] that parameterizes the unnormalized log density as $\log p_\theta(x) = f(x) + \log q_0(x)$, where $q_0(x)$ is a base Gaussian distribution and $f$ is a mixture of kernels. Specifically, $f(x) = \sum_{l=1}^{L} \alpha_l k(x, z_l)$, where $z_l$ are inducing points, $\alpha_l$ are the mixture weights, and $k(x, z_l)$ is the kernel function. The model can be trained with score-matching techniques with loss functions as SM, DSM or SSM. Alternatively, the model can use the

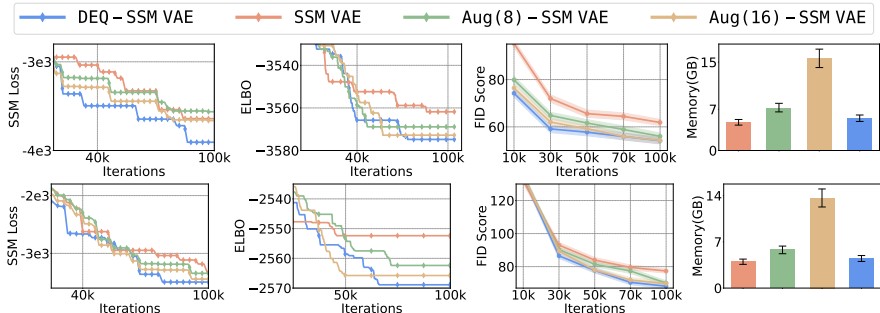

Figure 1: Comparing the performance of SSM VAE, Aug($k$)-SSM VAE, and DEQ-SSM VAE in terms of SSM loss, ELBO, FID score, and memory usage. First row: CelebA. Second row: Cifar10.

| Method | Dataset | FID↓ (10k iter) | FID↓ (50k iter) | FID↓ (100k iter) | #Param | Memory↓ | Training time/iter(s) |
|---|---|---|---|---|---|---|---|
| SSM VAE | | $95.23_{\pm0.50}$ | $65.59_{\pm0.52}$ | $61.85_{\pm0.55}$ | 8.8M | **4.0 GB** | 8.22 |
| Aug(8)-SSM VAE | CelebA | $79.87_{\pm0.61}$ | $61.62_{\pm0.61}$ | $56.02_{\pm0.57}$ | 15M | 7.3 GB | 37.57 |
| Aug(16)-SSM VAE | | $76.43_{\pm0.62}$ | $59.20_{\pm0.61}$ | $54.27_{\pm0.48}$ | 19M | 16 GB | 73.27 |
| **DEQ-SSM VAE** | | $\mathbf{74.21_{\pm0.52}}$ | $\mathbf{55.93_{\pm0.53}}$ | $\mathbf{54.17_{\pm0.48}}$ | 9.7M | 5.4 GB | 21.95 |
| SSM VAE | | $\mathbf{131.74_{\pm0.40}}$ | $83.93_{\pm0.42}$ | $77.35_{\pm0.38}$ | 7.1M | **3.8 GB** | **3.97** |
| Aug(8)-SSM VAE | Cifar10 | $135.28_{\pm0.40}$ | $81.41_{\pm0.35}$ | $70.05_{\pm0.37}$ | 13M | 6.3 GB | 17.45 |
| Aug(16)-SSM VAE | | $135.18_{\pm0.42}$ | $77.85_{\pm0.31}$ | $69.77_{\pm0.28}$ | 17M | 14 GB | 40.12 |
| **DEQ-SSM VAE** | | $134.24_{\pm0.39}$ | $\mathbf{77.72_{\pm0.33}}$ | $\mathbf{68.13_{\pm0.38}}$ | 7.5M | 4.4 GB | 9.98 |

Table 1: Performance comparison of models for image generation on CelebA and Cifar10: DEQ-SSM VAE leads in FID during training, except at 10k iterations where SSM VAE excels. DEQ-SSM VAE shows superior efficiency in parameters, memory, and time over increased depth.

SM loss but with curvature propagation (CP) to approximate the diagonal of the Hessian. We refer to these models as SM, DSM, SSM, and CP, respectively.

We consider the density estimation on two datasets: UCI(Parkinson/Redwine/Whitewine) [11] and high-dimensional Gaussian. The Gaussian datasets we estimate are 100 and 200 dimensional, respectively; each of the datasets is split into train, validation, and test sets with 4860, 600, and 540 samples, respectively. Each of the UCI datasets is split into train, validation, and test sets with 70%, 20% and 10%, respectively. In the experiment, we examine the performance of SM DKEF by increasing the number of layers of the kernel network such as $k$-Layer-SM DKEF ($k = 3, 8, 16$). We can also turn the sequence of MLP layers into a DEQ block termed as DEQ-SM DKEF.

We set the number of kernels to be 3. For the general score matching function, the feature extractor uses MLP with a different number of layers with hidden dimension 30 and sets Softplus as the activation function. For the DEQ variant, we turn the feature extractor into one DEQ block in the form of equation 5 with fully connected weights with dimension 30 and activation function Softplus. The models trained on the UCI and high-dimensional Gaussian datasets are the same except for the latent dimension which is 64 on UCI and 200 on Gaussian. The model is trained by Adam [17] for 200/300 epochs with batch size 32/128 on the high-dimensional Gaussian/UCI datasets, with learning rate $1e$-3, weight decay $1e$-12. We follow the procedure in [40], which trains the model using the score matching method with the objective function $\mathcal{J}_{\mathrm{SM}}$ introduced in Section 2.1.

### 4.2.1 UCI DATASETS: PARKINSON/REDWINE/WHITEWINE

Table 2 shows the results on the three UCI datasets. The results show that DEQ-SM DKEF outperforms the $k$-Layer-SM DKEF in terms of SSM loss and likelihood. Moreover, DEQ-SM DKEF is more efficient in terms of memory usage and training time compared to $k$-Layer-SM DKEF. The results also show that DEQ-SM DKEF outperforms DSM and CP in terms of SSM loss and likelihood.

| | Parkinson | | | | RedWine | | | | WhiteWine | | | |
|---|---|---|---|---|---|---|---|---|---|---|---|---|
| **Metrics** | SM Loss | Test LL | #Param | Mem | SM Loss | Test LL | #Param | Mem | SM Loss | Test LL | #Param | Mem |
| | (↓) | (↑) | | (↓) | (↓) | (↑) | | (↓) | (↓) | (↑) | | (↓) |
| DSM | $-71.2_{\pm1.2}$ | $-15.7_{\pm0.9}$ | 10k | 0.5 | $-24.8_{\pm3.0}$ | $-14.1_{\pm1.2}$ | 9k | 0.4 | $-17.8_{\pm4.9}$ | $-14.8_{\pm1.0}$ | 9k | 0.4 |
| CP | $-33.7_{\pm2.8}$ | $-16.9_{\pm1.3}$ | 10k | 0.7 | $-12.2_{\pm1.7}$ | $-15.3_{\pm1.5}$ | 9k | 0.6 | $-10.3_{\pm5.1}$ | $-16.1_{\pm0.5}$ | 9k | 0.6 |
| SSM | $-111.9_{\pm10.2}$ | $-15.1_{\pm0.9}$ | 10k | 0.5 | $-27.2_{\pm5.9}$ | $-13.8_{\pm0.7}$ | 9k | 0.5 | $-33.7_{\pm5.2}$ | $-14.8_{\pm0.8}$ | 9k | 0.5 |
| SSM-VR | $-121.1_{\pm7.9}$ | $-15.0_{\pm0.8}$ | 10k | 0.5 | $-27.2_{\pm5.9}$ | $-13.8_{\pm0.7}$ | 9k | 0.5 | $-33.7_{\pm5.2}$ | $-14.8_{\pm0.8}$ | 9k | 0.5 |
| DEQ-SMM DKEF* | $-143.2_{\pm8.9}$ | $-12.9_{\pm0.9}$ | 10k | 0.6 | $\mathbf{-42.9_{\pm8.9}}$ | $\mathbf{-12.9_{\pm0.9}}$ | 9k | 0.6 | $-43.7_{\pm6.3}$ | $-14.2_{\pm0.9}$ | 9k | 0.6 |
| SM | $-123.7_{\pm6.6}$ | $-15.1_{\pm0.7}$ | 10k | 1.2 | $-27.2_{\pm5.9}$ | $-13.8_{\pm0.7}$ | 9k | 1.0 | $-33.7_{\pm5.2}$ | $-14.8_{\pm0.8}$ | 9k | 1.0 |
| 8-Layer-SM DKEF | $-124.8_{\pm5.5}$ | $-15.3_{\pm0.9}$ | 26k | 2.1 | $-29.5_{\pm5.5}$ | $-13.5_{\pm0.8}$ | 25k | 1.6 | $-34.7_{\pm5.5}$ | $-14.4_{\pm0.5}$ | 25k | 1.6 |
| 16-Layer-SM DKEF | $-127.6_{\pm4.2}$ | $-13.4_{\pm0.6}$ | 51k | 3.6 | $-31.6_{\pm5.2}$ | $-12.7_{\pm0.8}$ | 50k | 2.7 | $-35.5_{\pm3.0}$ | $-14.5_{\pm0.6}$ | 50k | 2.7 |
| DEQ-SM DKEF* | $\mathbf{-152.1_{\pm4.7}}$ | $\mathbf{-12.4_{\pm0.7}}$ | 10k | 1.3 | $\mathbf{-44.5_{\pm4.1}}$ | $\mathbf{-12.2_{\pm0.7}}$ | 9k | 1.1 | $\mathbf{-45.01_{\pm4.7}}$ | $\mathbf{-13.9_{\pm0.5}}$ | 9k | 1.1 |

Table 2: Test SM loss and test log-likelihood of different models on the Parkinson, RedWine, and WhiteWine datasets. Mem stands for memory with unit GB.

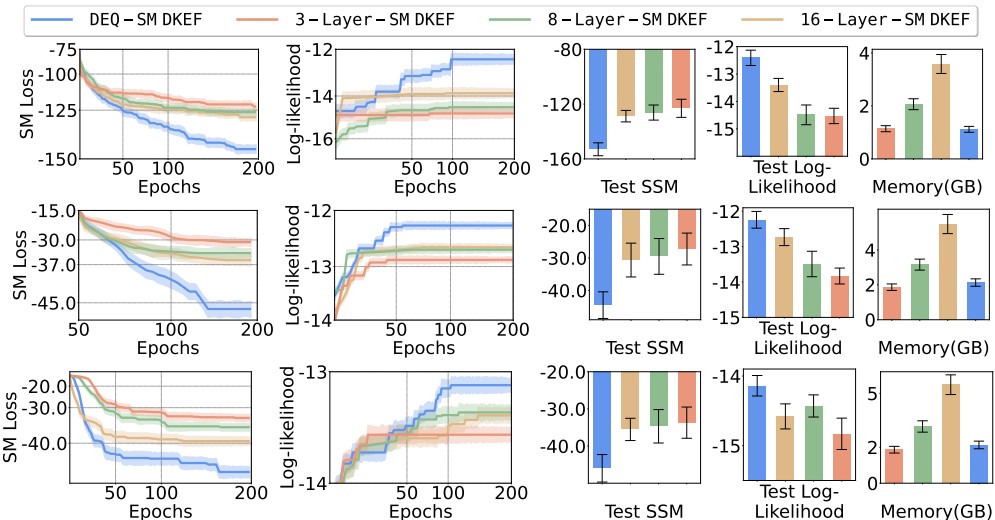

Figure 2: SSM/log-likelihood/memory usage of different DKEF models on Parkinson/RedWine/WhiteWine (Parkinson: row1; RedWine: row2; WhiteWine: row3) datasets.

### 4.2.2 HIGH-DIMENSIONAL GAUSSIAN

For the high-dimensional Gaussian datasets, the comparison of DEQ-SM DKEF, $k$-Layer-SM DKEF is shown in Table 3 and Figure 3 in terms of test loss, test log-likelihood, test Fisher divergence, and memory usage. The results show that the DEQ-based sliced score matching model can save memory while having the best performance on the test dataset compared to the $k$-Layer-SM DKEF model.

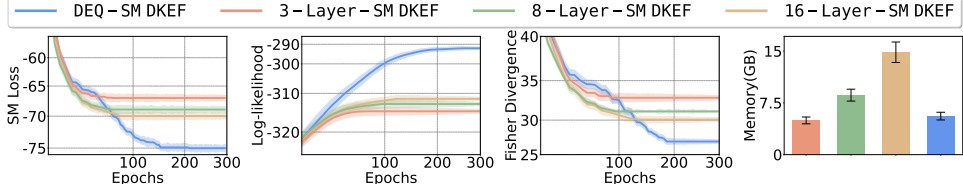

Figure 3: SM loss/log-likelihood/Fisher divergence on the test dataset, of the 200-dimensional Gaussian, from different DKEF methods. The memory used shows the memory for the training process.

| | Data Dim | Test SM Loss ↓ | Test LL ↑ | Test Fisher-Div ↓ | #Param | Memory |
|---|---|---|---|---|---|---|
| 3-Layer-SM DKEF | | -43.60 | -141.25 | 6.73 | 22k | 3.8 GB |
| 8-Layer-SM DKEF | 100- | -43.86 | -140.85 | **6.56** | 41k | 7.3 GB |
| 16-Layer-SM DKEF | dimension | -44.92 | -141.25 | 6.84 | 66k | 13.2 GB |
| **DEQ-SM DKEF** | | **-46.03** | **-138.86** | 6.72 | 22k | 3.9 GB |
| Ground Truth | | N/A | -137.14 | 0.0 | N/A | N/A |
| 3-Layer-SM DKEF | | -66.32 | -314.22 | 33.86 | 33k | 5.0 GB |
| 8-Layer-SM DKEF | 200- | -66.99 | -313.35 | 33.09 | 50k | 8.7 GB |
| 16-Layer-SM DKEF | dimension | -67.58 | -311.84 | 32.27 | 76k | 14.8 GB |
| **DEQ-SM DKEF** | | **-75.03** | **-292.56** | **25.42** | 33k | 5.6 GB |
| Ground Truth | | N/A | -283.79 | 0.0 | N/A | N/A |

Table 3: SM loss/log-likelihood/Fisher Divergence of different DKEF models for high-dimensional Gaussian.

### 4.3 NICE MODEL ON MNIST

NICE flow model [10] is another density estimate model. Unlike DKEF, its log-likelihood are tractable and can be directly trained with MLE. Following [40], we adopt NICE [10] as a sanity check for our DEQ-assisted score matching. We train it as well as a DEQ variant with SSM loss and compare it with the models – such as curvature propagation(CP) [25], DSM [43], ap-

| Metrics | Test SM Loss | Test LL | #Param | Mem Used |
|---|---|---|---|---|
| MLE | $-579$ | $-791$ | – | – |
| CP | $-1694$ | $-1517$ | – | – |
| DSM($\sigma = .1$) | $-3035$ | $-4363$ | – | – |
| DSM($\sigma = 1.74$) | $-97$ | $-8082$ | – | – |
| Approx BP | $-48$ | $-2288$ | – | – |
| SSM | $-2455_{\pm 52}$ | $-2058_{\pm 60}$ | 15M | 6.5 GB |
| DEQ-SSM | $-2548_{\pm 55}$ | $-1766_{\pm 48}$ | 14M | 5.5 GB |

Table 4: SM loss and log-likelihood for NICE Models on MNIST: the results of DSM, MLE, CP, and Approx BP results are from [40]; SSM and SSM DEQ results are averaged over three random seeds for memory usage.

proximate back-propagation (Approx BP) [19] – trained with SM loss, we also include maximum likelihood estimation (MLE) method as the likelihoods are tractable.

In this experiment, we consider the MNIST dataset [9], which contains 60K $28 \times 28$ grayscale images. We set the train, validation, and test datasets with 70%, 20%, and 10% of the whole dataset, respectively. We follow the setup in [40] and use it as a baseline model termed SSM NICE. At the core, SSM NICE contains 4 blocks with each containing 5 fully connected layers with 1000 hidden dimensions. Our DEQ-SSM NICE variant turns each of the blocks into a fully connected DEQ block of the form in equation 5 with 1000 hidden dimensions. Data are dequantized by adding uniform noise in the range $[-1/512, 1/512]$, and transformed using a logit transformation, $\log(x) - \log(1 - x)$. We train the model using Adam for 1000 iterations with a learning rate $1e$-3, weight decay $1e$-12, and batch size 128.

Table 4 compares the performance of DEQ-SSM against SSM and a few other baseline models, showing that DEQ-SSM outperforms SSM in both test SM loss and test log-likelihood. Meanwhile, DEQ-SSM takes less memory compared to SSM.

### 4.4 NCSN

We examine our DEQ integration in generative tasks using a score-based generative model called NCSN [37] with the training objective of SSM-VR. It is worth noting that our DEQ integration is orthogonal to the recent DEQ-based generative models [31], where DEQ is used to form the diffusion process in DDIM model [36], and its training objective does not involve derivatives. We follow the model setting in [37] whose core is a 4-cascaded RefineNet [23]. We use the SSM-VR as the objective function in this model and term the baseline model as SSM-VR NCSN. Our DEQ variant replaces the 4-cascaded RefineNet with two residue blocks with one of them turned into DEQ, we term this model as DEQ-SSM-VR NCSN. We also consider a variant of the model with two simple residue blocks, which we term as TwoRes-SSM-VR NCSN.

We train the model on the Cifar10 dataset [21] with 128 batch size for 200K iterations using Adam optimizer. For Adam, we set the learning rate as $1e$-4, with weight decay $1e$-12 and batch size 128. The results shown in Figure 4 and Table 5 compare the SSM-VR loss, FID score, and memory cost for retaining the computational graph of $\nabla_{\boldsymbol{x}} \log \tilde{p}_\theta(\boldsymbol{x})$ for backpropagation, as well as the overall memory cost. The results show that DEQ-SSM-VR NCSN achieves a significantly reduced memory footprint and faster computation while maintaining strong performance. In contrast, when the DEQ component is turned off in DEQ-SSM-VR NCSN, the resulting model TwoRes-SSM-VR NCSN is only marginally faster than its DEQ counterpart, but this came at the expense of notably diminished performance. In Figure 9 in the appendix, we provide samples generated by DEQ-SSM-VR NCSN.

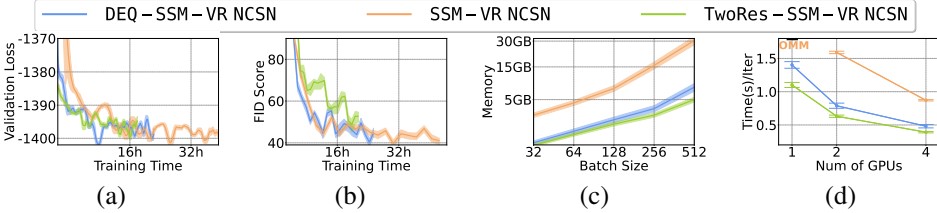

Figure 4: Performance comparison of NCSN models for Cifar10 generation: (a) Test loss vs. Training time, (b) FID score vs. Time, (c) Memory usage for computational graph retention in backpropagation, (d) Average iteration time vs. GPU count ('OMM' indicates 'out of memory').

## 5 CONCLUSION

In this work, we integrate DEQs into score matching models to address their memory constraints. We provide a convergence analysis of ap-

| Model | FID | Time(ms)/Iter | #Param | M. of $\nabla_{\boldsymbol{x}} \log \tilde{p}_\theta(\boldsymbol{x})$ | Tot Mem |
|---|---|---|---|---|---|
| SSM-VR NCSN | $40.31_{\pm 1.4}$ | $871_{\pm 10}$ | 28M | 7536 MiB | 40.4 GB |
| TwoRes-SSM-VR NCSN | $50.77_{\pm 1.5}$ | $391_{\pm 10}$ | 9M | 1701 MiB | 18.5 GB |
| DEQ-SSM-VR NCSN | $40.99_{\pm 1.0}$ | $481_{\pm 25}$ | 9M | 2104 MiB | 20.1 GB |

Table 5: Comparing NCSN-based image generation models on Cifar10: FID score, iteration time, and memory usage for computational graph and total training memory.

plying implicit differentiation to higher-order derivatives in the setting of DEQ. By strategically incorporating DEQs into core parts of existing models, we enhance depth and reduce memory requirements without compromising their performance. Our empirical experiments across various models, datasets, and tasks demonstrate that including DEQs in score matching not only significantly reduces memory usage but also improves computational efficiency and performance.

ACKNOWLEDGEMENT

This material is based on research sponsored by NSF grants DMS-2152762, DMS-2219956, and DMS-2208361 and DOE grant DE-SC0023490.

CONTRIBUTIONS STATEMENT

This paper started from a discussion between BW and AO in a workshop on using DEQ for computing Hessian. BW introduced the idea to QW and YH. QW explained DEQ and the phantom gradient technique to YH. YH implemented the whole idea, made the system work, and conducted all experiments. QW and YH wrote the first version of the paper focusing on non-experiments and experiments sections, respectively. All authors involved in writing the paper.

ETHICS STATEMENT

This paper introduces a memory-efficient approach for score matching using the deep equilibrium model. The proposed methods are validated in the classical benchmark problems, including density estimation and generative modeling. We do not see any potential ethical issues in our research.

REPRODUCIBILITY STATEMENT

To ensure reproducible research, we have made the following two major efforts: First, we include sufficient background materials and provide detailed mathematical derivation. Second, we submitted the code in the supplementary materials to ensure the experimental results can be easily reproduced.

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

## A APPENDIX

### A.1 MISSING PROOFS IN SECTION 3

In this section, we provide the missing proofs in Section 3.

*Proof of Proposition 1.* Since the function $f_{\boldsymbol{\theta}}(\boldsymbol{z}, \boldsymbol{x})$ is a contraction mapping, the Jacobian matrix $\left.\frac{\partial f_{\boldsymbol{\theta}}}{\partial \boldsymbol{z}}\right|_{\boldsymbol{z}^*}$ has its magitudes of the eigenvalues less than one. Therefore, the matrix $\boldsymbol{I} - \left.\frac{\partial f_{\boldsymbol{\theta}}}{\partial \boldsymbol{z}}\right|_{\boldsymbol{z}^*}$ is invertible. Then the implicit function theorem (e.g. Theorem 2.1 in Section 2 of [22]) implies that in a neighborhood of $\boldsymbol{z}^*$, $\boldsymbol{z}^*$ is a twice differentiable function of $\boldsymbol{x}$. Then we can differentiate through the fixed point equation $\boldsymbol{z}^* = f_{\boldsymbol{\theta}}(\boldsymbol{z}^*, \boldsymbol{x})$ to obtain

$$\frac{\partial \boldsymbol{z}^*}{\partial x_i} = \left.\frac{\partial f_{\boldsymbol{\theta}}}{\partial x_i}\right|_{\boldsymbol{z}^*} + \left.\frac{\partial f_{\boldsymbol{\theta}}}{\partial \boldsymbol{z}}\right|_{\boldsymbol{z}^*} \frac{\partial \boldsymbol{z}^*}{\partial x_i},$$

$$\frac{\partial \boldsymbol{z}^*}{\partial x_i} = \left(\boldsymbol{I} - \left.\frac{\partial f_{\boldsymbol{\theta}}}{\partial \boldsymbol{z}}\right|_{\boldsymbol{z}^*}\right)^{-1} \left.\frac{\partial f_{\boldsymbol{\theta}}}{\partial x_i}\right|_{\boldsymbol{z}^*}.$$

Similarly, we can differentiate through the fixed point equation $\boldsymbol{z}^* = f_{\boldsymbol{\theta}}(\boldsymbol{z}^*, \boldsymbol{x})$ twice to obtain

$$\frac{\partial^2 \boldsymbol{z}^*}{\partial x_i \partial x_j} = \left.\frac{\partial f_{\boldsymbol{\theta}}}{\partial \boldsymbol{z}}\right|_{\boldsymbol{z}^*} \frac{\partial^2 \boldsymbol{z}^*}{\partial x_i \partial x_j} + \left.\frac{\partial^2 f_{\boldsymbol{\theta}}}{\partial x_i \partial x_j}\right|_{\boldsymbol{z}^*} + \left.\frac{\partial^2 f_{\boldsymbol{\theta}}}{\partial \boldsymbol{z} \partial x_i}\right|_{\boldsymbol{z}^*} \frac{\partial \boldsymbol{z}^*}{\partial x_j}$$

$$+ \left.\frac{\partial^2 f_{\boldsymbol{\theta}}}{\partial \boldsymbol{z} \partial x_j}\right|_{\boldsymbol{z}^*} \frac{\partial \boldsymbol{z}^*}{\partial x_i} + \left.\frac{\partial^2 f_{\boldsymbol{\theta}}}{\partial \boldsymbol{z} \partial \boldsymbol{z}}\right|_{\boldsymbol{z}^*} \frac{\partial \boldsymbol{z}^*}{\partial x_i} \frac{\partial \boldsymbol{z}^*}{\partial x_j}$$

$$= \left(\boldsymbol{I} - \left.\frac{\partial f_{\boldsymbol{\theta}}}{\partial \boldsymbol{z}}\right|_{\boldsymbol{z}^*}\right)^{-1} \left(\left.\frac{\partial^2 f_{\boldsymbol{\theta}}}{\partial x_i \partial x_j}\right|_{\boldsymbol{z}^*} + \left.\frac{\partial^2 f_{\boldsymbol{\theta}}}{\partial \boldsymbol{z} \partial x_j}\right|_{\boldsymbol{z}^*} \frac{\partial \boldsymbol{z}^*}{\partial x_i}\right.$$

$$\left. + \left.\frac{\partial^2 f_{\boldsymbol{\theta}}}{\partial \boldsymbol{z} \partial x_i}\right|_{\boldsymbol{z}^*} \frac{\partial \boldsymbol{z}^*}{\partial x_j} + \left.\frac{\partial^2 f_{\boldsymbol{\theta}}}{\partial \boldsymbol{z} \partial \boldsymbol{z}}\right|_{\boldsymbol{z}^*} \frac{\partial \boldsymbol{z}^*}{\partial x_i} \frac{\partial \boldsymbol{z}^*}{\partial x_j}\right).$$

This concludes the proof. □

*Proof of Proposition 2.* We prove this result by induction on $K$. For $K = 1$, we have

$$\frac{\partial \boldsymbol{z}^{(1)}}{\partial x_i} = \left.\frac{\partial f_{\boldsymbol{\theta}}}{\partial x_i}\right|_{\boldsymbol{z}^{(0)}} + \left.\frac{\partial f_{\boldsymbol{\theta}}}{\partial \boldsymbol{z}}\right|_{\boldsymbol{z}^{(0)}} \frac{\partial \boldsymbol{z}^{(0)}}{\partial x_i} = \left.\frac{\partial f_{\boldsymbol{\theta}}}{\partial x_i}\right|_{\boldsymbol{z}^*}$$

This proves the base case. Now assume that the statement holds for $K - 1$, we have

$$\frac{\partial \boldsymbol{z}^{(K)}}{\partial x_i} = \left.\frac{\partial f_{\boldsymbol{\theta}}}{\partial x_i}\right|_{\boldsymbol{z}^{(K-1)}} + \left.\frac{\partial f_{\boldsymbol{\theta}}}{\partial \boldsymbol{z}}\right|_{\boldsymbol{z}^{(K-1)}} \frac{\partial \boldsymbol{z}^{(K-1)}}{\partial x_i}$$

$$= \left.\frac{\partial f_{\boldsymbol{\theta}}}{\partial x_i}\right|_{\boldsymbol{z}^*} + \left.\frac{\partial f_{\boldsymbol{\theta}}}{\partial \boldsymbol{z}}\right|_{\boldsymbol{z}^*} \left(\sum_{k=0}^{K-2} \left(\left.\frac{\partial f_{\boldsymbol{\theta}}}{\partial \boldsymbol{z}}\right|_{\boldsymbol{z}^*}\right)^k \left.\frac{\partial f_{\boldsymbol{\theta}}}{\partial x_i}\right|_{\boldsymbol{z}^*}\right)$$

$$= \sum_{k=0}^{K-1} \left(\left.\frac{\partial f_{\boldsymbol{\theta}}}{\partial \boldsymbol{z}}\right|_{\boldsymbol{z}^*}\right)^k \left.\frac{\partial f_{\boldsymbol{\theta}}}{\partial x_i}\right|_{\boldsymbol{z}^*}.$$

In particular, when $K \to \infty$, we have

$$\frac{\partial \boldsymbol{z}^*}{\partial x_i} = \sum_{k=0}^{\infty} \left(\left.\frac{\partial f_{\boldsymbol{\theta}}}{\partial \boldsymbol{z}}\right|_{\boldsymbol{z}^*}\right)^k \left.\frac{\partial f_{\boldsymbol{\theta}}}{\partial x_i}\right|_{\boldsymbol{z}^*} = \left(\boldsymbol{I} - \left.\frac{\partial f_{\boldsymbol{\theta}}}{\partial \boldsymbol{z}}\right|_{\boldsymbol{z}^*}\right)^{-1} \left.\frac{\partial f_{\boldsymbol{\theta}}}{\partial x_i}\right|_{\boldsymbol{z}^*}.$$

This concludes the proof of the first part.

Now we move to the second part, using the contraction of tensors, we will show that there is

$$\frac{\partial^2 \boldsymbol{z}^{(K)}}{\partial x_i \partial x_j} = \boldsymbol{A}_K \left.\frac{\partial^2 f_{\boldsymbol{\theta}}}{\partial x_i \partial x_j}\right|_{\boldsymbol{z}^*} + \boldsymbol{B}_K \left.\frac{\partial^2 f_{\boldsymbol{\theta}}}{\partial \boldsymbol{z} \partial x_j}\right|_{\boldsymbol{z}^*} \left.\frac{\partial f_{\boldsymbol{\theta}}}{\partial x_i}\right|_{\boldsymbol{z}^*}$$

$$+ \boldsymbol{B}_K \left.\frac{\partial^2 f_{\boldsymbol{\theta}}}{\partial \boldsymbol{z} \partial x_i}\right|_{\boldsymbol{z}^*} \left.\frac{\partial f_{\boldsymbol{\theta}}}{\partial x_j}\right|_{\boldsymbol{z}^*} + \boldsymbol{C}_K \left.\frac{\partial^2 f_{\boldsymbol{\theta}}}{\partial \boldsymbol{z} \partial \boldsymbol{z}}\right|_{\boldsymbol{z}^*} \left.\frac{\partial f_{\boldsymbol{\theta}}}{\partial x_i}\right|_{\boldsymbol{z}^*} \left.\frac{\partial f_{\boldsymbol{\theta}}}{\partial x_j}\right|_{\boldsymbol{z}^*}.$$

where the matrices $\boldsymbol{A}_K, \boldsymbol{B}_K, \boldsymbol{C}_K$ satisfy the following recursive equations:

$$\boldsymbol{A}_K = \left. \frac{\partial f_{\boldsymbol{\theta}}}{\partial \boldsymbol{z}} \right|_{\boldsymbol{z}^*} \boldsymbol{A}_{K-1} + \boldsymbol{I},$$

$$\boldsymbol{B}_K = \left. \frac{\partial f_{\boldsymbol{\theta}}}{\partial \boldsymbol{z}} \right|_{\boldsymbol{z}^*} \boldsymbol{B}_{K-1} + \sum_{k=0}^{K-2} \left( \left. \frac{\partial f_{\boldsymbol{\theta}}}{\partial \boldsymbol{z}} \right|_{\boldsymbol{z}^*} \right)^k,$$

$$\boldsymbol{C}_K = \left. \frac{\partial f_{\boldsymbol{\theta}}}{\partial \boldsymbol{z}} \right|_{\boldsymbol{z}^*} \boldsymbol{C}_{K-1} + \left( \sum_{k=0}^{K-2} \left( \left. \frac{\partial f_{\boldsymbol{\theta}}}{\partial \boldsymbol{z}} \right|_{\boldsymbol{z}^*} \right)^k \right) \left( \sum_{k=0}^{K-2} \left( \left. \frac{\partial f_{\boldsymbol{\theta}}}{\partial \boldsymbol{z}} \right|_{\boldsymbol{z}^*} \right)^k \right),$$

with initial conditions $\boldsymbol{A}_0 = \boldsymbol{I}, \boldsymbol{B}_0 = \boldsymbol{0}, \boldsymbol{C}_0 = \boldsymbol{0}$. We prove this by induction on $K$. For $K = 1$, we have

$$\frac{\partial^2 \boldsymbol{z}^{(1)}}{\partial x_i \partial x_j} = \left. \frac{\partial f_{\boldsymbol{\theta}}}{\partial \boldsymbol{z}} \right|_{\boldsymbol{z}^{(0)}} \frac{\partial^2 \boldsymbol{z}^{(0)}}{\partial x_i \partial x_j} + \left. \frac{\partial^2 f_{\boldsymbol{\theta}}}{\partial x_i \partial x_j} \right|_{\boldsymbol{z}^{(0)}} + \left. \frac{\partial^2 f_{\boldsymbol{\theta}}}{\partial \boldsymbol{z} \partial x_j} \right|_{\boldsymbol{z}^{(0)}} \frac{\partial \boldsymbol{z}^{(0)}}{\partial x_i} +$$

$$+ \left. \frac{\partial^2 f_{\boldsymbol{\theta}}}{\partial \boldsymbol{z} \partial x_i} \right|_{\boldsymbol{z}^{(0)}} \frac{\partial \boldsymbol{z}^{(0)}}{\partial x_j} + \left. \frac{\partial^2 f_{\boldsymbol{\theta}}}{\partial \boldsymbol{z} \partial \boldsymbol{z}} \right|_{\boldsymbol{z}^{(0)}} \frac{\partial \boldsymbol{z}^{(0)}}{\partial x_i} \frac{\partial \boldsymbol{z}^{(0)}}{\partial x_j}$$

$$= \left. \frac{\partial^2 f_{\boldsymbol{\theta}}}{\partial x_i \partial x_j} \right|_{\boldsymbol{z}^*}$$

This verifies the base case.

Now assume that the statement holds for $K - 1$, we have

$$\frac{\partial^2 \boldsymbol{z}^{(K)}}{\partial x_i \partial x_j} = \left. \frac{\partial f_{\boldsymbol{\theta}}}{\partial \boldsymbol{z}} \right|_{\boldsymbol{z}^{(K-1)}} \frac{\partial^2 \boldsymbol{z}^{(K-1)}}{\partial x_i \partial x_j} + \left. \frac{\partial^2 f_{\boldsymbol{\theta}}}{\partial x_i \partial x_j} \right|_{\boldsymbol{z}^{(K-1)}} + \left. \frac{\partial^2 f_{\boldsymbol{\theta}}}{\partial \boldsymbol{z} \partial x_j} \right|_{\boldsymbol{z}^{(K-1)}} \frac{\partial \boldsymbol{z}^{(K-1)}}{\partial x_i} +$$

$$+ \left. \frac{\partial^2 f_{\boldsymbol{\theta}}}{\partial \boldsymbol{z} \partial x_i} \right|_{\boldsymbol{z}^{(K-1)}} \frac{\partial \boldsymbol{z}^{(K-1)}}{\partial x_j} + \left. \frac{\partial^2 f_{\boldsymbol{\theta}}}{\partial \boldsymbol{z} \partial \boldsymbol{z}} \right|_{\boldsymbol{z}^{(K-1)}} \frac{\partial \boldsymbol{z}^{(K-1)}}{\partial x_i} \frac{\partial \boldsymbol{z}^{(K-1)}}{\partial x_j}$$

$$= \left. \frac{\partial f_{\boldsymbol{\theta}}}{\partial \boldsymbol{z}} \right|_{\boldsymbol{z}^{(K-1)}} \left( \boldsymbol{A}_K \left. \frac{\partial^2 f_{\boldsymbol{\theta}}}{\partial x_i \partial x_j} \right|_{\boldsymbol{z}^{(K-1)}} + \boldsymbol{B}_K \left. \frac{\partial^2 f_{\boldsymbol{\theta}}}{\partial \boldsymbol{z} \partial x_j} \right|_{\boldsymbol{z}^{(K-1)}} \frac{\partial f_{\boldsymbol{\theta}}}{\partial x_i} \right.$$

$$\left. + \boldsymbol{B}_K \left. \frac{\partial^2 f_{\boldsymbol{\theta}}}{\partial \boldsymbol{z} \partial x_i} \right|_{\boldsymbol{z}^{(K-1)}} \frac{\partial f_{\boldsymbol{\theta}}}{\partial x_j} + \boldsymbol{C}_K \left. \frac{\partial^2 f_{\boldsymbol{\theta}}}{\partial \boldsymbol{z} \partial \boldsymbol{z}} \right|_{\boldsymbol{z}^{(K-1)}} \frac{\partial f_{\boldsymbol{\theta}}}{\partial x_i} \frac{\partial f_{\boldsymbol{\theta}}}{\partial x_j} \right)$$

$$+ \left. \frac{\partial^2 f_{\boldsymbol{\theta}}}{\partial x_i \partial x_j} \right|_{\boldsymbol{z}^*} + \left. \frac{\partial^2 f_{\boldsymbol{\theta}}}{\partial \boldsymbol{z} \partial x_j} \right|_{\boldsymbol{z}^*} \left( \sum_{k=0}^{K-2} \left( \left. \frac{\partial f_{\boldsymbol{\theta}}}{\partial \boldsymbol{z}} \right|_{\boldsymbol{z}^{(K-1)}} \right)^k \left. \frac{\partial f_{\boldsymbol{\theta}}}{\partial x_i} \right|_{\boldsymbol{z}^{(K-1)}} \right)$$

$$+ \left. \frac{\partial^2 f_{\boldsymbol{\theta}}}{\partial \boldsymbol{z} \partial x_i} \right|_{\boldsymbol{z}^*} \left( \sum_{k=0}^{K-2} \left( \left. \frac{\partial f_{\boldsymbol{\theta}}}{\partial \boldsymbol{z}} \right|_{\boldsymbol{z}^{(K-1)}} \right)^k \left. \frac{\partial f_{\boldsymbol{\theta}}}{\partial x_j} \right|_{\boldsymbol{z}^{(K-1)}} \right)$$

$$+ \left. \frac{\partial^2 f_{\boldsymbol{\theta}}}{\partial \boldsymbol{z} \partial \boldsymbol{z}} \right|_{\boldsymbol{z}^*} \left( \sum_{k=0}^{K-2} \left( \left. \frac{\partial f_{\boldsymbol{\theta}}}{\partial \boldsymbol{z}} \right|_{\boldsymbol{z}^{(K-1)}} \right)^k \left. \frac{\partial f_{\boldsymbol{\theta}}}{\partial x_i} \right|_{\boldsymbol{z}^{(K-1)}} \right) \left( \sum_{k=0}^{K-2} \left( \left. \frac{\partial f_{\boldsymbol{\theta}}}{\partial \boldsymbol{z}} \right|_{\boldsymbol{z}^{(K-1)}} \right)^k \left. \frac{\partial f_{\boldsymbol{\theta}}}{\partial x_j} \right|_{\boldsymbol{z}^{(K-1)}} \right)$$

By collecting terms and using the fact that the value of $\boldsymbol{z}^{(K-1)}$ equals $\boldsymbol{z}^*$, we verify that the matrices $\boldsymbol{A}_K, \boldsymbol{B}_K, \boldsymbol{C}_K$ satisfy their respective recursive equations. Since the matrices $\left. \frac{\partial f_{\boldsymbol{\theta}}}{\partial \boldsymbol{z}} \right|_{\boldsymbol{z}^*}$ is a contraction mapping, the recursive equations imply that the matrices $\boldsymbol{A}_K, \boldsymbol{B}_K, \boldsymbol{C}_K$ converge to the following limits:

$$\lim_{K \to \infty} \boldsymbol{A}_K = \left( \boldsymbol{I} - \left. \frac{\partial f_{\boldsymbol{\theta}}}{\partial \boldsymbol{z}} \right|_{\boldsymbol{z}^*} \right)^{-1},$$

$$\lim_{K \to \infty} \boldsymbol{B}_K = \left( \boldsymbol{I} - \left. \frac{\partial f_{\boldsymbol{\theta}}}{\partial \boldsymbol{z}} \right|_{\boldsymbol{z}^*} \right)^{-2},$$

$$\lim_{K \to \infty} \boldsymbol{C}_K = \left( \boldsymbol{I} - \left. \frac{\partial f_{\boldsymbol{\theta}}}{\partial \boldsymbol{z}} \right|_{\boldsymbol{z}^*} \right)^{-3}.$$

Hence, we have the following limit:

$$\lim_{K \to \infty} \frac{\partial^2 \boldsymbol{z}^{(K)}}{\partial x_i \partial x_j} = \left( \boldsymbol{I} - \frac{\partial f_{\boldsymbol{\theta}}}{\partial \boldsymbol{z}} \bigg|_{\boldsymbol{z}^*} \right)^{-1} \left( \frac{\partial^2 f_{\boldsymbol{\theta}}}{\partial x_i \partial x_j} \bigg|_{\boldsymbol{z}^*} + \frac{\partial^2 f_{\boldsymbol{\theta}}}{\partial \boldsymbol{z} \partial x_j} \bigg|_{\boldsymbol{z}^*} \frac{\partial \boldsymbol{z}^*}{\partial x_i} \right.$$
$$\left. + \frac{\partial^2 f_{\boldsymbol{\theta}}}{\partial \boldsymbol{z} \partial x_i} \bigg|_{\boldsymbol{z}^*} \frac{\partial \boldsymbol{z}^*}{\partial x_j} + \frac{\partial^2 f_{\boldsymbol{\theta}}}{\partial \boldsymbol{z} \partial \boldsymbol{z}} \bigg|_{\boldsymbol{z}^*} \frac{\partial \boldsymbol{z}^*}{\partial x_i} \frac{\partial \boldsymbol{z}^*}{\partial x_j} \right)$$
$$= \frac{\partial^2 \boldsymbol{z}^*}{\partial x_i \partial x_j}.$$

This concludes the proof. □

## B IMPLEMENTATION OF DEQ

---
**Algorithm 1** Implementation of the fixed point iteration in PyTorch-style pseudocode.

---
1: **procedure** FIXEDPOINTITERATION($f_{\boldsymbol{\theta}}, \boldsymbol{x}, K$)
2:     **with** torch.no_grad():
3:         $\boldsymbol{z}^* = \text{RootSolver}(f_{\boldsymbol{\theta}}, \boldsymbol{x})$
4:     $\boldsymbol{z}^{(0)} = \boldsymbol{z}^*$
5:     **for** $k = 0$ to $K - 1$ **do**
6:         $\boldsymbol{z}^{(k+1)} = f_{\boldsymbol{\theta}}(\boldsymbol{z}^{(k)}, \boldsymbol{x})$
7:     **end for**
8:     **return** $\boldsymbol{z}^{(K)}$
9: **end procedure**

---

## C EMPIRICAL ANALYSIS OF PHANTOM GRADIENT ERRORS

In our DEQ integration, we use a simple implementation of the phantom gradient with $K = 2$ and without any additional damping. This is because we find that this simple implementation is sufficient for the DEQ integration to work well in practice. In this section, we provide additional empirical analysis of the phantom gradient error in the DEQ model. We follow the setting of the experiments in the DKEF model in Section 4.2 and use the Pakinson dataset for the experiments. We train the model with SM and SSM loss respectively and at each training step, we compute the score function using the exact gradient for the DEQ block and the phantom gradient with $K = 2$. We also compute the loss gradient with respect to the parameters used in the two cases.

### C.1 ANALYSIS OF ERRORS IN THE SCORE NORM

We first consider the score norm error, which is defined as the relative error of the scores obtained from outputs of the DEQ with $K = 2$ using phantom gradient and DEQ with true gradient computed through implicit differentiation. We use $\boldsymbol{s}$ to denote the score from the exact gradient for the DEQ block and $\tilde{\boldsymbol{s}}$ to denote the score from the phantom gradient with $K = 2$. The relative error is defined as

$$err = \frac{||\boldsymbol{s} - \tilde{\boldsymbol{s}}||}{||\boldsymbol{s}||} * 100\%.$$

In Figure 5, we show the distribution of the score norm error from training the same model on the Pakinson dataset with SM and SSM losses, respectively. In both cases, the score norm error is small, which indicates that the phantom gradient is a good approximation of the exact gradient. Additionally, in Figure 6, we show the corresponding distribution of the score cosine similarity. The results demonstrate that the scores obtained from the exact gradient and the phantom gradient with $K = 2$ are highly correlated.

### C.2 ANALYSIS ON THE PARAMETER GRADIENT COSINE SIMILARITY

In this section, we consider how the error stems from the phantom gradient affects the optimization process. To this end, we consider the cosine similarity of the gradient of the loss function with

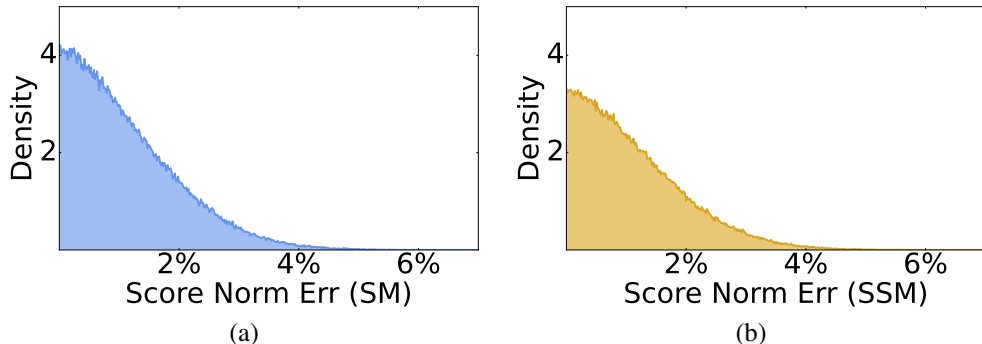

Figure 5: (a) and (b) show the distribution of the score norm error from SM and SSM, respectively. The relative error concentrates around zero, indicating that the phantom gradient with $K = 2$ is a good approximation of the exact gradient.

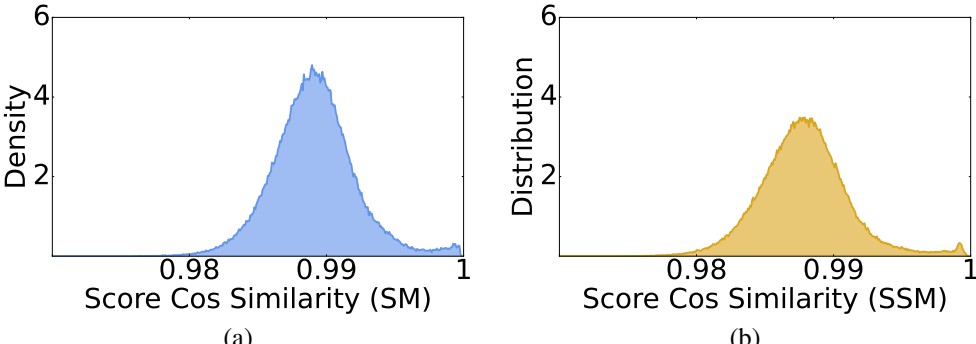

Figure 6: (a) and (b) show the distribution of the score cosine similarity from SM and SSM respectively. The scores obtained from the exact gradient and the phantom gradient with $K = 2$ are highly correlated.

respect to the parameters from the exact gradient in the DEQ block and the phantom gradient with $K = 2$. The result is shown in Figure 7. The results show that the cosine similarity is close to one, which further indicates that the phantom gradient is a good approximation of the exact gradient.

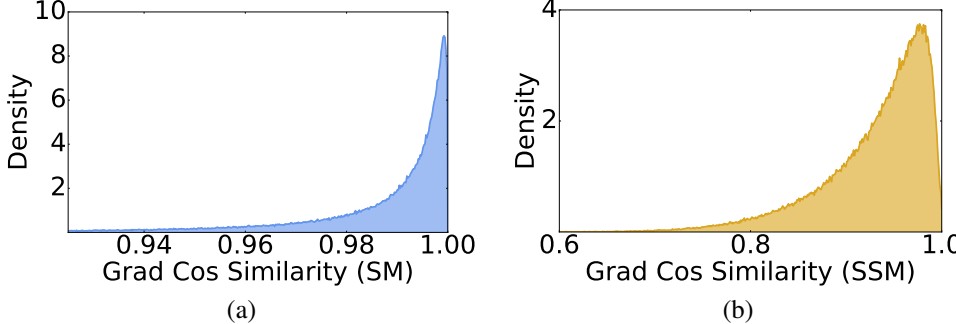

Figure 7: (a) and (b) show the distribution of the cosine similarity of the gradient of the loss function with respect to the parameters from the exact gradient in the DEQ block and the phantom gradient with $K = 2$ from SM and SSM, respectively. The cosine similarity shows that the loss gradient from the phantom gradient is highly correlated with the loss gradient from the exact gradient.

# D  ADDITIONAL DETAILS ON EXPERIMENTS

## D.1  DKEF MODEL

The deep kernel exponential families (DKEF) model parameterizes the unnormalized density as $\log p_{\boldsymbol{\theta}}(\boldsymbol{x}) = f(\boldsymbol{x}) + \log q_0(\boldsymbol{x})$, where $q_0(\boldsymbol{x})$ is a base distribution and $f(\boldsymbol{x})$ is a deep kernel exponential family. In our implementation, the deep kernel exponential family is defined as

$$f(\boldsymbol{x}) = \sum_{k=l}^{3} \alpha_l k(\boldsymbol{x}, \boldsymbol{z}_l)$$

where $\boldsymbol{z}_l$ are induced points, $\alpha_l$ are mixture weights. The kernel function $k(\boldsymbol{x}, \boldsymbol{z}_l)$ is defined as

$$k(\boldsymbol{x}, \boldsymbol{z}_l) = \sum_{r=1}^{3} \rho_r \exp\left(-\frac{1}{2\sigma_r^2}\|\phi_r(\boldsymbol{x}) - \phi_r(\boldsymbol{z}_l)\|^2\right).$$

Here, $\phi_r(\cdot)$ is the $r$-th feature extractor, $\rho_r$ is the mixture coefficient, and $\sigma_r$ is the length scale. We use the DEQ layer to model the feature extractor $\phi_r(\cdot)$ of the form equation 5 with fully connected weights and Softplus activation function. Additionally, $g(\boldsymbol{y})$ in this case is a single layer MLP also with Softplus nonlinearity.

## D.2  NICE MODEL

Nonlinear independent components estimation (NICE) is a flow-based density estimate model that has four coupling layers with each layer containing five dense hidden linear layers with Softplus activation function. The DEQ model is used to replace the five dense hidden linear layers in each coupling layer to be of the form equation 5 with Softplus activation function. The function $g(\boldsymbol{y})$ in this case is a single layer MLP with Softplus nonlinearity.

## D.3  VAE MODEL

We include the architecture used in the VAE model in Table 6. The model is based on the model in [40] with the addition of the [Augmented Pre-processing Layer] in the score estimator. The specific configuration of the [Augmented Pre-processing Layer] is shown in Table 7.

## D.4  NCSN MODEL

Noise conditional score network (NCSN) [37; 38] is a score-based generative model that learns the score function from data and then uses the score function to generate samples. The score function is parameterized as a neural network. In our implementation, the DEQ residual block contains two components that form a map as $\boldsymbol{z} \rightarrow h(\boldsymbol{z}) + g(\boldsymbol{y})$ where $h$ is a (normalized) residue block with each convolution weight has its Frobuinus norm constrained to ensure contractiveness and $g(\boldsymbol{y})$ is a ResBlock that transforms the input data, as in the original NCSN implementation [37], See Table 8 and Table 9 for details.

# E  SAMPLED IMAGES GENERATED BY DEQ-ASSISTED GENERATIVE MODELS

In this section, we provide the sampled images generated by DEQ-assisted models in Section 4. This includes the images generated by DEQ-SSM VAE model on the CelebA dataset (Figure 8) and the images generated by DEQ-SSM based NCSN model on the Cifar10 dataset (Figure 9).

# F  CONVERGENCE OF THE FIXED-POINT SOLVER

In our experiments, we utilize the simple fixed point iteration (Picard iteration) to solve the fixed point of the DEQ layer. This is because we aimed to attribute differences directly to DEQ integration, not solver efficiency. In this section, we provide the convergence results of DEQ-based models in

| Name | Configuration |
|------|---------------|
| Implicit Encoder | $5 \times 5$ conv; $m$ channels; stride $2 \times 2$; padding 2; ReLU |
| | $5 \times 5$ conv; $2m$ channels; stride $2 \times 2$; padding 2; ReLU |
| | $5 \times 5$ conv; $4m$ channels; stride $2 \times 2$; padding 2; ReLU |
| | $5 \times 5$ conv; $8m$ channels; stride $2 \times 2$; padding 2; ReLU |
| | 512 Dense; ReLU |
| | $D_{\boldsymbol{z}}$ Dense |
| Decoder | Dense; ReLU |
| | $5 \times 5$ conv$^{\top}$; $4m$ channels; stride $2 \times 2$; padding 2; out padding 1; ReLU |
| | $5 \times 5$ conv$^{\top}$; $2m$ channels; stride $2 \times 2$; padding 2; out padding 1; ReLU |
| | $5 \times 5$ conv$^{\top}$; $m$ channels; stride $2 \times 2$; padding 2; out padding 1; ReLU |
| | $5 \times 5$ conv$^{\top}$; $c$ channels; stride $2 \times 2$; padding 2; out padding 1; Tanh |
| Score Estimator | Concat$[\boldsymbol{x}$, Softplus(Dense($\boldsymbol{z}$))$]$ |
| | [Augmented Pre-processing Layer] |
| | $5 \times 5$ conv; $m$ channels; stride $2 \times 2$; padding 2; Softplus |
| | $5 \times 5$ conv; $2m$ channels; stride $2 \times 2$; padding 2; Softplus |
| | $5 \times 5$ conv; $4m$ channels; stride $2 \times 2$; padding 2; Softplus |
| | $5 \times 5$ conv; $8m$ channels; stride $2 \times 2$; padding 2; Softplus |
| | 512 Dense; Softplus |
| | $D_{\boldsymbol{z}}$ Dense |

Table 6: The basic model of each part of the VAE model used in Section 4.1. $m$ is a hyper-parameter to be set to define the number of channels of each convolution hidden layer. $c$ denotes the number of channels of the original image data. $D_{\boldsymbol{z}}$ is the dimension of the latent space. The [Augmented Pre-processing Layer] contains layers in addition to the base model in [40] that augments the input data. The specific configuration of these layers is shown in Table 7.

| Model | Specification |
|-------|---------------|
| Sequential CNN | concat$[\boldsymbol{x}$, Softplus(Dense($\boldsymbol{z}$))$]$ |
| | 1: $5 \times 5$ conv; $m$ channels; stride $1 \times 1$; padding 2; Softplus |
| ($n$-layers) | 2: $5 \times 5$ conv; $m$ channels; stride $1 \times 1$; padding 2; Softplus |
| | .... |
| | $n$ : $5 \times 5$ conv; $m$ channels; stride $1 \times 1$; padding 2; Softplus |
| DEQ | concat$[\boldsymbol{x}$, Softplus(Dense($\boldsymbol{z}$))$]$ |
| | $f(\boldsymbol{z}) :=$ Softplus$(\boldsymbol{W}\boldsymbol{z} + g(\boldsymbol{y}))$ |

Table 7: Specifications of the [Augmented Pre-processing Layer] in the score estimator of the VAE model 6. The input transformation $g(\boldsymbol{y})$ in the DEQ shares the same architecture as one layer of the Sequential CNN. Meanwhile, the weight $\boldsymbol{W}$ is the linearized version of such a convolutional layer.

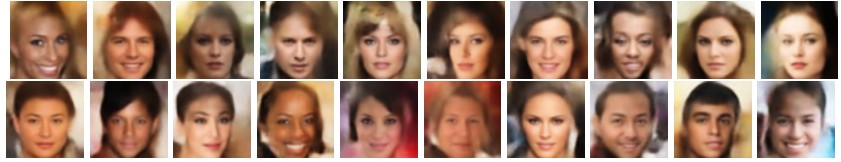

Figure 8: Images generated by DEQ-SSM VAE model on the CelebA dataset.

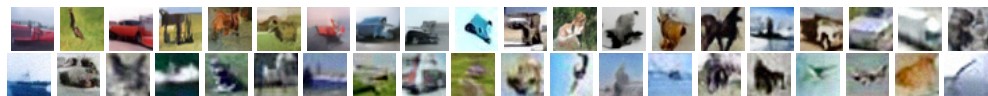

Figure 9: Images generated by DEQ-SSM based NCSN model on the Cifar10 dataset.

Section 4. In Table 10, we present the numerical tolerance using absolute error in $L_2$-norm between two consecutive iterations, the maximum number of iterations, and the averaged number of iterations observed in the experiments.

| SSM-VR NCSN |
|---|
| $3 \times 3$ Conv2D,  $N_C$: $3 \to 128$ |
| ResBlock,  $N_C$: $128 \to 128$ |
| ResBlock,  $N_C$: $128 \to 256$ |
| ResBlock down,  $N_C$: $256 \to 256$ |
| ResBlock,  $N_C$: $256 \to 256$ |
| ResBlock down,  $N_C$: $256 \to 256$ |
| ResBlock,  $N_C$: $256 \to 256$ |
| ResBlock down,  $N_C$: $256 \to 256$ |
| ResBlock,  $N_C$: $256 \to 256$ |
| RefineBlock,  $N_C$: $256 \to 256$ |
| RefineBlock,  $N_C$: $256 \to 128$ |
| RefineBlock,  $N_C$: $128 \to 128$ |
| RefineBlock,  $N_C$: $128 \to 128$ |
| 3x3 Conv2D,  $N_C$: $128 \to 3$ |

(a)

| DEQ-SSM-VR NCSN |
|---|
| $3 \times 3$ Conv2D,  $N_C$: $3 \to 128$ |
| ResBlock,  $N_C$: $128 \to 256$ |
| **DEQ-ResBlock,**  $N_C$: $256 \to 256$ |
| ResBlock,  $N_C$: $256 \to 256$ |
| ResBlock down,  $N_C$: $256 \to 256$ |
| ResBlock,  $N_C$: $256 \to 256$ |
| RefineBlock,  $N_C$: $256 \to 256$ |
| RefineBlock,  $N_C$: $256 \to 128$ |
| 3x3 Conv2D,  $N_C$: $128 \to 3$ |

(b)

Table 8: The architectures of SSM-VR NCSN (a) vs. DEQ-SSM-VR NCSN (b). $N_C$ denotes the number of channels.

| DEQ-ResBlock |
|---|
| $f(\boldsymbol{z}) := h(\boldsymbol{z}) + g(\boldsymbol{y})$, |
| where $h(\boldsymbol{z})$ is a ResBlock(normalized) |
| with $N_C$: $256 \to 256$ |

| $g(\boldsymbol{y})$ |
|---|
| $3 \times 3$ Conv2D,  $N_C$: $3 \to 128$ |
| ResBlock,  $N_C$: $256 \to 256$ |

Table 9: The architecture of DEQ-ResBlock in the DEQ-SSM-VR NCSN. $g(\boldsymbol{y})$ is the first CNN layer and the first ResBlock in (b) of Table 8 that transforms the input data to the hidden variable. $N_C$ denotes the number of channels.

| Model | Absolute Error Error | Max Iterations | Average Iterations |
|---|---|---|---|
| DEQ-SM DKEF | $5e$-6 | 128 | 12.8 |
| DEQ-SSM DKEF | $5e$-6 | 128 | 14.5 |
| DEQ-SSM VAE (CelebA) | $1e$-5 | 128 | 18.9 |
| DEQ-SSM VAE (Cifar10) | $1e$-5 | 128 | 15.2 |
| DEQ-SSM NICE | $5e$-6 | 128 | 11.8 |
| DEQ-SSM NCSN | $1e$-5 | 128 | 16.6 |

Table 10: List of convergence-related parameters, including absolute error, the maximum number of iterations, and the average number of iterations observed during experiments. The results show that by constraining the Frobenius norm of the linear map in the DEQ layer as well as the small contractive factor near zero of the Softplus activation function and also reusing the learned fixed point from the previous forward pass, we often obtain good convergence results.

