# OpenReview forum: "Efficient Score Matching with Deep Equilibrium Layers"
_ICLR.cc/2024/Conference — ICLR 2024 poster_

### Official Review · Reviewer_YwTh · 2023-10-30

**Soundness:** 3 good
**Presentation:** 3 good
**Contribution:** 2 fair
**Rating:** 6
**Confidence:** 4

**Summary:**

The paper proposes a novel score-matching method that uses deep equilibrium networks (DEQs) to overcome the memory bottleneck usually associated to score-matching.

The main contributions of the paper are the proposed method involving DEQs, a theoretical analysis involving implicit differentiation for higher-order derivatives, and an empirical validation that compares the DEQ model with some other methods for solving score-matching problems, like sliced score-matching.

The theoretical analysis consists of two propositions. The first proposition is concerned with computing the first and second derivatives of the solution mapping of the fixed point equation used in the DEQ and these formulas involve inverting a matrix (typical for implicit differentiation methods). The second proposition is focused on approximating the derivatives of the solution mapping formulated in Proposition 1 without inverting any matrices, by using a truncated Neumann series approximation.

**Edit: after the rebuttal by the authors, the assumptions on the types of activation functions considered have been strengthened and my doubts about the theoretical contributions have been addressed. I therefore raise my soundness score and overall score to a 6.**

The empirical validation is split into four parts, each centered around a task that can be solved by score-matching methods. In each of these tasks the frugality with respect to memory is validated without showing loss in accuracy - the plots suggest that the most accurate methods in terms of various metrics (Frechet Inception Distance, slice score-matching loss, ELBO, etc) is often the DEQ method, although it is not often the fastest method nor is it always the most memory frugal.

**Strengths:**

The novelty and empirical validation of the claims in the paper seem worthwhile. I don't know of any methods that combine DEQs with score matching, although DEQs and the phantom gradient method of section 3 are already well-known and studied in the papers that introduced them. It is also clear from the numerical experiments that this performs well empirically and that the memory overhead is not so large, especially compared to augmented sliced score matching methods.

**Weaknesses:**

The theoretical parts of the paper are not correct.

For instance in proposition 1 it is required that the function f_theta is continuously differentiable; this is not the case for the ReLU and will necessitate a non-smooth implicit function theorem (see, e.g., [Bolte 2021]). There even appears the second derivative of f_theta but this is not compatible with the stated assumptions.

Then in Proposition 2 there is no justification that z* is going to be twice differentiable (and in the examples using ReLU, it might not even be differentiable). If f_theta is continuously differentiable then one can use the implicit function theorem to show that z* is continuously differentiable in a nieghborhood (assuming an invertibility condition holds on the partial derivatives, which will as you are assuming a contractive mapping) but this does not give twice differentiability of the solution map z*.

I find these to be major gaps in the theoretical analysis that do not appear to be trivial or easily fixed.

Small inconsistency: It is written “Given most activation functions, such as ReLU and Softplus used in equation 5 are contractive,” but this is not correct, the ReLU is not contractive (its Lipschitz constant is 1).

"Nonsmooth Implicit Differentiation for Machine-Learning and Optimization" - Jérôme Bolte, Tam Le, Edouard Pauwels, Tony Silveti-Falls, NeurIPS 2021.

**Questions:**

Are there any conditions you can impose on f_theta that will ensure that z* is twice differentiable that don't also rule out using non-smooth functions like ReLU in the DEQ?

DEQs and the phantom gradient method in section 3 are already well-known, what is the contribution here besides plugging them into score-matching?

---

> ### Author Response · Authors · 2023-11-17
> **Response to Reviewer YwTh**
>
> Thank you for your thoughtful review and valuable feedback. We respectfully disagree with the comment that the theoretical parts of the paper are not correct. In what follows, we provide some detailed clarification to address your concerns.
>
> ---
>
> **Q1. In proposition 1 it is required that the function $f_\theta$ is continuously differentiable; this is not the case for the ReLU and will necessitate a non-smooth implicit function theorem (see, e.g., [Bolte 2021]). There even appears the second derivative of f_theta but this is not compatible with the stated assumptions.**
>
> **Reply:** Thank you for raising the concern about the regularity of function $f_\theta$. In Proposition 1, we have now specified that $f_\theta$ is twice continuously differentiable.
> This choice aligns with the general regularity assumption of the score-based methods; see e.g. [Hyärinen 2005, Song 2020]. In our experiments, we opted for the smooth function like Softplus instead of ReLU -- following existing benchmarks -- to ensure the required twice differentiability, especially in the context of employing SM (score matching) loss.
>
> It's worth noting that some other works on general DEQ models employ ReLU, and therefore, theoretical results like those in [Bolte 2021] are crucial. To address this, we have referenced this work in our literature review of DEQ.
>
> [Bolte 2021] Bolte et al. Nonsmooth Implicit Differentiation for Machine-Learning and Optimization, NeurIPS 2021.
>
> [Hyärinen 2005] Aapo Hyärinen. Estimation of non-normalized statistical models by score matching. JMLR, 6(4), 2005.
>
> [Song 2020] Song et al. Sliced score matching: A scalable approach to density and score estimation. UAI 2020.
>
> ---
>
> **Q2. Proposition 2: there is no justification that $z^\ast$ is going to be twice differentiable (and in the examples using ReLU, it might not even be differentiable). If f_theta is continuously differentiable then one can use the implicit function theorem to show that $z^\ast$ is continuously differentiable in a neighborhood (assuming an invertibility condition holds on the partial derivatives, which will as you are assuming a contractive mapping) but this does not give twice differentiability of the solution map $z^\ast$.**
>
> **Reply:** We have updated our revision to clarify that $f_\theta$ is twice continuously differentiable, which aligns with the practice and guarantees that $z^\ast$ is twice differentiable in a neighborhood of $z^\ast$; see e.g. Theorem 2.1 in Section 2 of  [Lang 1983].
>
> [Lang 1983] Serge Lang, Real Analysis. Addison-Wesley, 1983.
>
> ---
>
> **Q3. It is written “Given most activation functions, such as ReLU and Softplus used in equation 5 are contractive,” but this is not correct, the ReLU is not contractive (its Lipschitz constant is 1).**
>
> **Reply:** Indeed, ReLU is merely non-expansive, but since the linear function within the activation is already a contraction, then $f_\theta$ will be a contraction as well. In our revision, we removed the mention of ReLU since it is not used in our experiments and in general score matching-based models; see e.g. [Hyärinen 2005, Song 2020].
>
> [Hyärinen 2005] Aapo Hyärinen. Estimation of non-normalized statistical models by score matching. JMLR, 6(4), 2005.
>
> [Song 2020] Song et al. Sliced score matching: A scalable approach to density and score estimation. UAI 2020.
>
> ---
>
> **Q4. Are there any conditions you can impose on f_theta that will ensure that $z^\ast$ is twice differentiable that don't also rule out using non-smooth functions like ReLU in the DEQ?**
>
> **Reply:** Extending the work in [Bolte 2021] to accommodate higher-order derivatives for non-smooth functions such as ReLU is intriguing, but it goes beyond the scope of our current study. While DEQ models in other studies often incorporate ReLU activations, our approach, following [Song 2020], employs smooth functions like Softplus when twice differentiability is needed.
>
> ---
>
> **Q5. DEQs and the phantom gradient method in section 3 are already well-known, what is the contribution here besides plugging them into score-matching?**
>
> **Reply**: Our approach stands out by integrating DEQs into score matching, allowing for the effective handling of higher-order derivatives and the memory bottleneck of score matching. As a result, we have observed notable enhancements in empirical performance. To the best of our knowledge, our work is the first one that studies DEQ in score matching that requires the computation of higher derivatives.
>
>
> ---
>
> We have updated our submission based on the reviewer's feedback, with the revision highlighted in blue. We are happy to address further questions on our paper. Thank you for considering our rebuttal.

---

> ### Comment · Reviewer_YwTh · 2023-11-19
> **Response to rebuttal**
>
> Thank you for your comprehensive response. With the assumptions strengthened and clearly stated to require that the activation function is twice continuously differentiable, I agree that the theoretical claims are salvageable. Although this excludes the ReLU activation, if the softmax is the targeted activation for this application then I agree that this is not too strict of an assumption. I am still unconvinced regarding point 5. For these reasons I have increased my overall score.

---

> > ### Author Response · Authors · 2023-11-19
> > **Further Response to Reviewer YwTh**
> >
> > Thank you for considering our rebuttal and we appreciate your further feedback and your endorsement of our submission.
> >
> > Regarding your further concern about point 5, i.e. “What is the contribution besides plugging DEQs and phantom gradient method into score-matching?” Our primary contribution lies in addressing a crucial class of problems in score matching-based density estimation or generative models that involve derivatives in the loss function, leading to higher-order derivatives during backpropagation. To tackle this challenge, we have integrated DEQ layers into score matching and examined the limits of second-order derivatives using the phantom gradient method for the first time. Though the approach we use is not very complicated, this approach has empirically shown significant enhancements in model performance, particularly in handling complex derivative computations efficiently.
> >
> > Thank you again for considering our rebuttal and further feedback.

---

### Official Review · Reviewer_hmh8 · 2023-10-31

**Soundness:** 3 good
**Presentation:** 3 good
**Contribution:** 3 good
**Rating:** 8
**Confidence:** 4

**Summary:**

**Edit: Score increased after rebuttal.**

This paper proposes the use of deep equilibrium layers (DEQs) for the task of score-matching. The motivation for doing this is that score-matching techniques are typically expensive (in both time and memory) as they require computing gradients and/or Hessians of the model, due to the need to store the entire computational graph in memory for backprop. In contrast, DEQ methods do not need to store the entire computational graph. In Section 3, Propositions 1 and 2 give results on several derivatives needed to train score-based models with DEQ layers. Section 4 provides a fairly thorough empirical evaluation of the proposed methods across a variety of tasks, including density estimation and generative modeling. The proposed methods generally show comparable or better performance than the baselines, with low memory costs.

**Strengths:**

- Score-matching techniques are a fairly hot topic, and this paper aims to solve an important issue with these models, namely their high memory costs.
- The experimental methodology is generally sound, and the provided evidence generally supports the claims made throughout the paper.
- The proposed method shows consistently favorable performance in comparison with the considered baselines, while still retaining lower memory costs.
- To the best of my knowledge, this is the first work to use DEQs in the context of score-matching for density estimation tasks.

**Weaknesses:**

- The main weakness of the paper is that the technical contribution of the paper is somewhat limited. In particular, the authors apply standard DEQ modeling techniques to the task of score-matching. While this combination is novel, the techniques are a relatively straightforward application of existing DEQ methodology. Moreover, DEQ models in the context of diffusion models has been previously explored [2].
- It was unclear in some places what was novel in this work and what has been derived previously. For instance, in Proposition 1, Equation 8 has certainly already appeared in the DEQ literature (Theorem 1 of [1]), but it was unclear to me if the second-order results (Equation 9) was novel. Similarly, Proposition 2 (at least in part) appears in [3], and it was not entirely clear to me what aspects of this were novel.
- The results in Section 4.2.1 were a little confusing to me.
     - What exactly are DSM, CP, and SSM referring to in Table 2? My understanding is that these rows correspond to some version of the DKEF model fit with these score-matching techniques, but the details were unclear.
     - Moreover, the acronym "CP" is used here before it is introduced later in the paper.
     - I would have also expected Table 2 to include memory usage (similar to Table 3) as this is one of the key motivations for the proposed method.
     - The description of DKEF itself was also unclear -- what is $q_0$? Where are the learnable parameters $\theta$ appearing?
- Given that there exists previous work on DEQ diffusion models [2], I might have expected a comparison with the methods in [2] in Section 4.4, or at least some rationale for not doing so.

### Minor Comments
- In Section 4.1, "encoder" should be $q_\phi(z | x)$ and "decoder" should be $p_\theta(x | z)$
- Typo in the caption of Figure 3: "momory" should be "memory"

In general, I found this paper to be well-motivated with thorough and convincing empirical results, but perhaps limited in its novelty.

### References

[1] [Bai et al., Deep Equilibrium Models](https://proceedings.neurips.cc/paper_files/paper/2019/file/01386bd6d8e091c2ab4c7c7de644d37b-Paper.pdf)

[2] [Pokle et al., Deep Equilibrium Approaches to Diffusion Models](https://arxiv.org/pdf/2210.12867.pdf)

[3] [Geng et al., On Training Implicit Models](https://arxiv.org/pdf/2111.05177.pdf)

**Questions:**

See weakness section.

---

> ### Author Response · Authors · 2023-11-17
> **Response to Reviewer hmh8**
>
> Thank you for your thoughtful review and valuable feedback. In what follows, we provide point-by-point responses to your comments.
>
> ---
>
> **Q1: The main weakness of the paper is that the technical contribution of the paper is somewhat limited. In particular, the authors apply standard DEQ modeling techniques to the task of score-matching. While this combination is novel, the techniques are a relatively straightforward application of existing DEQ methodology. Moreover, DEQ models in the context of diffusion models has been previously explored [2].**
>
> **Reply:** Our primary contribution lies in identifying and addressing a specific class of problems in score matching-based density estimation or generative models that involve derivatives in the loss function, leading to higher-order derivatives during backpropagation. To tackle this challenge, we have integrated DEQ layers into score matching and examined the limits of second-order derivatives using the phantom gradient method. This approach has empirically shown significant enhancements in model performance, particularly in handling complex derivative computations efficiently.
>
> Regarding the reference to DEQ models in diffusion models (as per [2]), it's important to note that while [2] does involve DEQ and diffusion models, its focus is distinctly different from ours. The work in [2] reformulates the entire diffusion process of the DDIM model as a DEQ, and their loss function does not involve derivatives.
>
> ---
>
> **Q2: It was unclear in some places what was novel in this work and what has been derived previously. For instance, in Proposition 1, Equation 8 has certainly already appeared in the DEQ literature (Theorem 1 of [1]), but it was unclear to me if the second-order results (Equation 9) was novel. Similarly, Proposition 2 (at least in part) appears in [3], and it was not entirely clear to me what aspects of this were novel.**
>
> **Reply:** As far as we know, the second-order results are new in the literature.
>
> ---
>
> **Q3: What exactly are DSM, CP, and SSM referring to in Table 2? My understanding is that these rows correspond to some version of the DKEF model fit with these score-matching techniques, but the details were unclear.**
>
> **Reply:** DKEF is a score-based density estimation model. The DSM(denoising score matching), CP(curvature propagation), and SSM(sliced score matching) variants have the same model architecture but with different ways of computing the loss. The DSM and SSM compute DSM loss and SSM loss, respectively, and CP computes an approximated, complex-valued diagonal of Hessian. Thank you for pointing these out, and we have updated the revision to include relevant discussion.
>
> ---
>
> **Q4: Moreover, the acronym "CP" is used here before it is introduced later in the paper.**
>
> **Reply:** Thank you for pointing this out. CP stands for curvature propagation, and we have clarified this in the revision.
>
> ---
>
> **Q5: I would have also expected Table 2 to include memory usage (similar to Table 3) as this is one of the key motivations for the proposed method.**
>
> **Reply:** Thank you for the suggestion. We have reported memory usage in Table 2 of the revised paper.
>
> ---
>
> **Q6: The description of DKEF itself was also unclear -- what is $q_0$? Where are the learnable parameters $\theta$ appearing?**
>
>
> **Reply:** $q_0$ is the log density of a mean zero Gaussian distribution with learnable variance. The main learnable parameters in DKEF occur in the modeling of the kernel function $K$.
>
> ---
>
> **Q7: Given that there exists previous work on DEQ diffusion models [2], I might have expected a comparison with the methods in [2] in Section 4.4, or at least some rationale for not doing so.**
>
> **Reply:** We want to clarify that the approach taken in [2] focuses on reformulating the diffusion process of the DDIM diffusion model as a DEQ model, where the loss function does not involve derivatives. This contrasts with our paper, where we utilize a score-based diffusion model (NCSN) and employ sliced score matching (SSM) loss for training, which inherently involves derivatives. Recognizing the significance of this distinction, we have added a clarification in Section 4 of our paper.
>
> ---
>
> **Q8: Typo.**
>
> **Reply:** Thank you for pointing them out. In the revised paper, we have corrected them and further improved the presentation.
>
> ---
>
> We have updated our submission based on the reviewer's feedback, with the revision highlighted in blue. We are happy to address further questions on our paper. Thank you for considering our rebuttal.

---

> > ### Comment · Reviewer_hmh8 · 2023-11-20
> >
> > Thanks for the detailed responses, particularly with regards to Q1. This has addressed my concerns. I still find the key contribution of the work (Prop 1 and Prop 2) fairly incremental, being a straightforward extension of the existing first-order results. However, the results are sound, with a fair amount of empirical evidence to bolster the claims, and I'm willing to increase my score.
> >
> > Regarding Q2, I think it would strengthen the clarity of the paper to make this point more explicit still (space permitting).

---

> > > ### Author Response · Authors · 2023-11-20
> > > **Further Response to Reviewer hmh8**
> > >
> > > Thank you for considering our rebuttal and we appreciate your further feedback and your endorsement of our submission. We will revise our paper according to your suggestion.

---

### Official Review · Reviewer_CgBT · 2023-10-31

**Soundness:** 3 good
**Presentation:** 2 fair
**Contribution:** 2 fair
**Rating:** 6
**Confidence:** 4

**Summary:**

Training models with score matching objective can be computationally expensive as it involves computation of higher-order derivatives like Hessian. Therefore, alternate objectives like sliced score matching (SSM) and denoising score matching (DSM) were proposed.

This work proposes to use deep equilibrium model (DEQ)-based architectures to train networks with different score matching objectives in order to reduce the memory requirements. DEQs use implicit gradients which reduce the memory footprint of backward pass as it does not require creation of an explicit computation graph for all the layers of the network. However, computing implicit gradients for DEQs need  matrix inversion. This work uses phantom gradients to circumvent matrix inversion.

The benefits of using DEQ-based architecture for SSM and DSM has been shown on different density estimation tasks with NICE and deep kernel exponential family (DKEF) models. Further, benefits of using this architecture is also shown for generative modeling with VAEs and NCSN. Overall, the experiments show that using DEQs leads to lower ELBO and test SSM loss, as well as reduced memory requirements.

**Strengths:**

As indicated by results in Figure 1-5 and Tables 1-5, DEQs outperform or match the performance of non-weight tied architectures for optimizing with SSM objectives on density estimation tasks involving NICE and DKEF architectures, as well as on score estimation for VAEs and NCSN. Further, they consistently have lower memory footprint. The plots also seem to indicate that DEQ converge faster and have lower training and test loss, while matching or outperforming perceptual quality of generated images in terms of FID.

**Weaknesses:**

- The paper is very sparse on various implementation details of DEQ. Some of the important details that are missing are:
    - The choice of fixed point solver is unclear. From the code shared in the supplementary, it seems like fixed point iterations were used. However, there are other fixed point solver methods (e.g. Newton methods, Anderson acceleration, Jacobi, Gauss-Siedel etc.) that can converge faster, and it is unclear why simple fixed point iterations were preferred over these other faster methods. I presume that memory footprint could be one of the reasons. The reasoning behind various design and implementation should be explicitly stated. Further, for each task, the number of fixed point iterations should be stated as this is an important hyper parameter.
    - The original work on phantom gradients (Geng et al. 2022) implements these gradients as a convex combination with parameter $\lambda$ i.e. $z^\star = \lambda f(z^\star, x) + (1 - \lambda) z^\star$. However, it seems that this paper doesn’t follow this as per Algorithm 1. Is there an empirical evidence of optimality for the choice of  $\lambda=1$? Further, the paper also skips on values of hyper parameter $K$ used for phantom gradients for different tasks and datasets.
    - The paper is very sparse on implementation details of different network architectures. In general, choice of g(y) in Equation 5 is unclear for different model architectures for instance in Table 7. Similarly, DEQ architectural details, especially choice of g(y) are missing for other architectures.
- There is limited empirical analysis on the nature of convergence for different architectures. We don’t know how convergent the fixed points are and if there are additional benefits of using additional test-time iterations of the DEQ architecture to squeeze improved test-time performance. It will useful to report the final values of the absolute difference between consecutive iterates of DEQ $\|| z^{t} - f(z^{t}, x)\||_2$ or relative difference measure as $\frac{\|| f(z^t, x) - z^t \||_2} {\|| z^t\||_2}$. Even reporting the final values of these quantities at $z^\star$ is useful.
- In the discussion in Section 3.2, it is important to point that most of the practical DEQs are not globally convergent — neither uniqueness not existence of the fixed point is guaranteed--- and thus they are not well-posed. This applies to the DEQs used in this work as well. Further, it is possible to converge to limit cycles as noted in [2][3]. This is the motivation behind deep equilibrium models like MonDEQs [1] which have explicit guarantees on uniqueness and existence of a fixed point.
- This paper derives closed form expressions for second order implicit gradients used for backprop in DEQ, but empirical analysis of stability and performance of these second order implicit gradients is missing.
- Scalability analysis for score matching is missing. As DEQs have constant memory requirements, these models should be more scalable compared to non-weight tied models as the dimension of data increases. It would be useful to recreate a figure similar to Figure 2 in Song et al. 2020 [4].
- I am not sure if a fair comparison is being made between non weight-tied and DEQ models across all the tasks. Ideally, one should equalize either for the number of parameters or FLOPs. However, these numbers are not included for all the tasks/results in the experiments section.

[1] Winston, Ezra, and J. Zico Kolter. "Monotone operator equilibrium networks." Advances in neural information processing systems 33 (2020): 10718-10728.

[2] Bai, Shaojie, J. Zico Kolter, and Vladlen Koltun. "Deep equilibrium models." Advances in Neural Information Processing Systems 32 (2019).

[3] Anil, Cem, et al. "Path Independent Equilibrium Models Can Better Exploit Test-Time Computation." Advances in Neural Information Processing Systems 35 (2022): 7796-7809.

[4] Song, Yang, et al. "Sliced score matching: A scalable approach to density and score estimation." Uncertainty in Artificial Intelligence. PMLR, 2020.

**Questions:**

1. Could the authors clarify which tasks in the experiments sections use a DEQ model that needs computation of Hessian? As far as I can tell, only DKEF trained on UCI datasets uses the original score matching loss. Besides this, all other tasks use sliced score matching (SSM) loss or denoising score matching loss, both of which do require computation of Hessians. In that case, could the authors state the memory requirements of the networks in Table 2. Also, could the authors add these numbers for SSM and SSM with variance reduction (SSM-VR) in Table 2?
2. There are several missing details in the tables of experimental sections:
    1. The size of models in terms of number of parameters should be indicated in Tables 2, 4, and 5.
    2. The implementation details of DEQ-SSM NICE state — “Our DEQ-DDM NICE variant turns each of the blocks into a fully connected DEQ block of the form in equation 5 with 1000 hidden dimensions.” It is unclear what g(y) is in this case.
    3. Please include the size of CelebA images used in Section 4.1

---

> ### Author Response · Authors · 2023-11-17
> **Response to Reviewer CgBT (1/3)**
>
> Thank you for your thoughtful review and valuable feedback. In what follows, we provide point-by-point responses to your comments.
>
> ---
>
> **Q1: The choice of fixed point solver is unclear. From the code shared in the supplementary, it seems like fixed point iterations were used. However, there are other fixed point solver methods (e.g. Newton methods, Anderson acceleration, Jacobi, Gauss-Siedel etc.) that can converge faster, and it is unclear why simple fixed point iterations were preferred over these other faster methods. I presume that memory footprint could be one of the reasons. The reasoning behind various designs and implementations should be explicitly stated. Further, for each task, the number of fixed point iterations should be stated as this is an important hyperparameter.**
>
>
> **Reply:** We use a simple fixed-point solver, Picard iteration, to compute the fixed point in the DEQ model. Our focus in this paper is to show that using DEQ can significantly improve memory efficiency in score matching. Exploring the benefits of faster fixed-point solvers can be a future work. We aim to attribute differences directly to DEQ integration rather than solver efficiency. This rationale and the number of iterations used for each task have been clarified in Appendix F of the revised paper.
>
> ---
>
> **Q2: The original work on phantom gradients (Geng et al. 2022) implements these gradients as a convex combination with parameter $\lambda$. However, it seems that this paper doesn’t follow this as per Algorithm 1. Is there empirical evidence of optimality for the choice of $\lambda$? Further, the paper also skips on values of hyperparameter $K$ used for phantom gradients for different tasks and datasets.**
>
> **Reply:** In our implementation of phantom gradients, we chose $\lambda =1$ and $K=2$ for all tasks and datasets, not necessarily because they are optimal, but for simplicity and consistency. This approach allowed us to make fair comparisons with previous models without introducing additional hyperparameters that could complicate the analysis. We recognize the importance of these details and have included this explanation in our paper to clarify our choices and their rationale; see Appendix C for details.
>
> ---
>
> **Q3: The paper is very sparse on implementation details of different network architectures. In general, the choice of $g(y)$ in Equation 5 is unclear for different model architectures for instance in Table 7. Similarly, DEQ architectural details, especially choice of $g(y)$ are missing for other architectures.**
>
>
> **Reply:** We have updated the details about the choices of $g(y)$ in Appendix D. The choice of $g(y)$ follows the original design of the models, often being either single-layer MLP or single-layer convolutional network. In the NCSN case, we use residual block following the original NCSN implementation.
>
> ---
>
>
> **Q4: There is limited empirical analysis on the nature of convergence for different architectures. We don’t know how convergent the fixed points are and if there are additional benefits of using additional test-time iterations of the DEQ architecture to squeeze improved test-time performance. It will be useful to report the final values of the absolute difference between consecutive iterates of DEQ $||z^{t} - f(z^{t}, x)||_2$ or relative difference measure as $\frac{||f(z^t,x)-z^t||_2}{||z^t||_2}$. Even reporting the final values of these quantities at $z^\ast$ is useful.**
>
> **Reply:** Thank you for highlighting the potential enhancements in DEQ-based models by increasing iterations at test time. In our experiments, we opted for a simple tolerance threshold with a maximum number of iterations, consistent for both training and testing phases for simplicity. In the revision, we have included convergence-related hyperparameters in our experiments, now detailed in Appendix F. The tolerance is set to $5e-6$ or $1e-5$, and the absolute difference between consecutive iterate $||z^{t}-f(z^{t},x)||_2$ is employed. We observe that the DEQ maintains good convergence in the experiments, possibly due to the constraint on the Frobenius norm of the linear part of the DEQ layer and the small contractive factor near zero of the Softplus activation function.
>
> ---

---

> ### Author Response · Authors · 2023-11-17
> **Response to Reviewer CgBT (2/3)**
>
> **Q5: In the discussion in Section 3.2, it is important to point that most of the practical DEQs are not globally convergent — neither uniqueness not existence of the fixed point is guaranteed--- and thus they are not well-posed. This applies to the DEQs used in this work as well. Further, it is possible to converge to limit cycles as noted in [2][3]. This is the motivation behind deep equilibrium models like MonDEQs [1] which have explicit guarantees on uniqueness and existence of a fixed point.**
>
> **Reply:** We appreciate your insights regarding the convergence properties of DEQs. In our implementation, we have ensured the existence and uniqueness of the fixed point by constraining the Frobenius norm of the weight matrix to be less than 1 and selecting non-expansive activation functions, making the DEQ layer contractive. While this approach is more straightforward compared to MonDEQs, it effectively guarantees both existence and uniqueness. Incorporating MonDEQ, with its explicit guarantees on these aspects, certainly presents an interesting direction for future work.
>
> ---
>
> **Q6: This paper derives closed form expressions for second order implicit gradients used for backprop in DEQ, but empirical analysis of stability and performance of these second order implicit gradients is missing.**
>
> **Reply:** In the revision, specifically in Appendix C, we present a detailed empirical analysis aimed at evaluating the effectiveness of using $K=2$ for phantom gradients. This analysis includes a direct comparison between the phantom gradient approach (with $K=2$) and the exact gradient method for the DEQ block.
>
> We found that the score function remains largely consistent when employing phantom gradients with $K=2$, indicating that the approximation does not significantly deviate from the exact gradient in terms of both the values and trends. Moreover, we observe a strong correlation in the parameter update direction between the phantom gradient method and the exact gradient approach. This observation suggests that even with $K=2$, our method maintains a high degree of fidelity to the exact gradient.
>
> ---
>
> **Q7: Scalability analysis for score matching is missing. As DEQs have constant memory requirements, these models should be more scalable compared to non-weight tied models as the dimension of data increases. It would be useful to recreate a figure similar to Figure 2 in Song et al. 2020 [4].**
>
> **Reply:** Our work concentrates on model depth scalability in score matching-based models rather than data dimensionality. It's crucial to note that the challenge of quadratic dependency on data dimensionality stems from the requirement to store second-order derivatives, which is inherent to the score matching (SM) loss function itself. This particular dimensionality bottleneck is effectively addressed by variants like sliced score matching (SSM), which reduces the computational burden of high-dimensional data.
>
> Regarding the benefit of constant memory in DEQs, the results in Tables 1, 2, and 3 show that DEQ-based models outperform baseline models and their increased depth counterparts while maintaining low memory footprints.
>
> ---
>
> **Q8: I am not sure if a fair comparison is being made between non weight-tied and DEQ models across all the tasks. Ideally, one should equalize either for the number of parameters or FLOPs. However, these numbers are not included for all the tasks/results in the experiments section.**
>
> **Reply:** We have included the parameter count for each experiment. Utilizing the parameter efficiency of DEQ models, our DEQ-based models often have a similar parameter count as the models chosen as baselines but with much-improved performance.
>
> ---
>
> **Q9: Could the authors clarify which tasks in the experiments sections use a DEQ model that needs computation of Hessian? As far as I can tell, only DKEF trained on UCI datasets uses the original score matching loss. Besides this, all other tasks use sliced score matching (SSM) loss or denoising score matching loss, both of which do not require the computation of Hessians. In that case, could the authors state the memory requirements of the networks in Table 2. Also, could the authors add these numbers for SSM and SSM with variance reduction (SSM-VR) in Table 2?**
>
> **Reply:**  Indeed, only DKEF on Gaussian and UCI datasets uses SM loss and hence computing Hessian is needed to form the loss. However, we want to emphasize that even though SSM loss and DSM loss only involves first-order derivative in the loss, higher-order derivatives still occur during backpropagation for updating the weights.
>
> We have updated Table 2 to include memory consumption and performance of SSM and SSM-VR.
>
> ---

---

> ### Author Response · Authors · 2023-11-17
> **Response to Reviewer CgBT (3/3)**
>
> **Q10: The size of models in terms of number of parameters should be indicated in Tables 2, 4, and 5.**
>
> **Reply:** We have added the parameter count in these Tables.
>
> ---
>
> **Q11: The implementation details of DEQ-SSM NICE state — “Our DEQ-DDM NICE variant turns each of the blocks into a fully connected DEQ block of the form in equation 5 with 1000 hidden dimensions.” It is unclear what $g(y)$ is in this case.**
>
> **Reply:** $g(y)$, in this case, is a simple MLP transform of the input with $y\rightarrow \sigma(Wy+b)$, where $\sigma$ is Softplus and $W, b$ are learnable weights.
>
> ---
>
> **Q12: Please include the size of CelebA images used in Section 4.1**
>
> **Reply:** The size of each image in CelebA is $64\times 64\times 3$, and we have added the description in our revision.
>
>
> ---
>
> We have updated our submission based on the reviewer's feedback, with the revision highlighted in blue. We are happy to address further questions on our paper. Thank you for considering our rebuttal.

---

> > ### Comment · Reviewer_CgBT · 2023-11-21
> >
> > Thank you for your detailed feedback and for additional details on training of DEQs, hyperparameters and architecture. It is mentioned that the Frobenius norm of the weight matrix is constrained to be less than 1 during training. It is unclear to me how is this is implemented in practice in context of DEQs in this work, e.g. via clipping weight matrices after each gradient descent update, projected gradient descent, regularization etc. Further, were other architectures ( e.g. architectures like Aug(8/16)-SSM VAE, 8/16-Layer-SM DKEF etc.) trained in a similar way i.e. Frobenius norm of weights was constrained to be less than 1? What is the overhead in training time of DEQs used here compared to the other architectures? Finally, does time/iteration in Table 1 correspond to training time or inference time?

---

> ### Author Response · Authors · 2023-11-21
>
> Thank you for the opportunity to discuss these aspects of our work in more detail. In the following, we address each of your concerns.
>
> 1. **Implementation of Frobenius Norm Constraint in DEQs:**
>    In our DEQ-based models, the Frobenius norm constraint for the weight $\bf{W}$ is implemented using a "normalization" step $ \frac{\bf{W}}{\lambda (|\bf{W}| +\epsilon)}$ to ensure the weight in fixed-point iteration is constrained. Here, $ \lambda $ is a hyperparameter that is greater than 1, and $ \epsilon $ is a small positive number added for numerical stability.
>
> 2. **Training of Non-DEQ architectures:**
>    We did not apply the Frobenius norm constraint for other architectures in our study, such as Aug(8/16)-SSM VAE and 8/16-Layer-SM DKEF. This decision allowed us to adhere to the original model settings and retain fully unconstrained weights for the non-DEQ architectures.
>
> 3. **Training Overhead:**
>    The Frobenius norm constraint in DEQs only involves a few operations for computing the Frobenius norm, divisions, and summations, and thus does not result in a noticeable overhead in training time.
>
> 4. **Time/Iteration in Table 1:**
>    The time/iteration metric presented in Table 1 refers to the training time. These results indicate that DEQ-SSM models are more time-efficient in training than models with increased depth. This demonstrates the efficiency of DEQ-based models in backpropagation when loss functions contain derivatives.
>
> We hope this response adequately addresses your questions and further clarifies our work. In light of your feedback, we have revised our manuscript accordingly to clarify these aspects.

---

### Official Review · Reviewer_H3Py · 2023-10-31

**Soundness:** 3 good
**Presentation:** 2 fair
**Contribution:** 3 good
**Rating:** 6
**Confidence:** 3

**Summary:**

Score matching methods attempt to learn probability distributions by computing a loss on the derivative of the log likelihoods, thereby circumventing the need to compute the normalising constant. Unfortunately, they are time and particularly memory intensive, on account of requiring to compute the Hessian during backpropagation. This paper uses deep equilibrium models (DEQs) to improve time and memory consumption. Experiments show that with the same computational and memory budget, DEQ models can outperform non-DEQ models.

**Strengths:**

This is overall a good quality paper.
- The paper cites an appropriate amount of related literature and places itself well within the literature.
- The use of phantom differentiation is nice (mentioned in section 3.4). Compute the forward iterations of the FP solver with auto-diff off, then turn it on and run K more steps.
- Experiments are convincing and broad, and show that DEQ models consume less memory and achieve better predictive performance than some existing methods. (Contribution +)
- The paper is moistly clearly written. The illustrations are clear and informative. Tables are presented well. (However I do have questions about section 3.1, as below, hence the low "Presentation" score). (Presentation ++)
- The mathematics and formulation are mostly easy to understand, and they appear to be okay. (Soundness ++)
- This paper solves an important problem of speeding up training in score-based methods (Contribution ++)

**Weaknesses:**

I am willing to raise my score if the authors can respond to my queries below.

- My biggest concern is related to my first Question in the box below. Why is this method actually principled? As far as I understand, a DEQ model is used as a kind of surrogate for the true score function, and then the fixed point of this surrogate is plugged into the score function and the Fisher Divergence is minimised. Does this result in the DEQ model being somehow close to the true score function, and in particular, the fixed point of the DEQ becomes close to the fixed point of the score function? I would appreciate it if the authors could spend some more time describing the method itself (i.e., I feel section 3.1 is too short). (Presentation --)
- Proposition 2. It seems to require $K \to \infty$, however the authors state they use $K=2$. There is no quantification of the error in finite $K$ settings. Furthermore, it is not clear how error in the derivatives translates to errors in the score, or errors in the Fisher divergence. (Soundness -)

Minor:
- The first sentence in the abstract doesn't seem to make sense, however I could be wrong. Double check with someone else. I would write "Score matching methods, *which* estimate probability densities without computing the normalization constant, are particularly useful in deep learning"
- I believe there is an incorrect statement in section 3.2. "The well-posedness of the fixed-point equation 5 can be guaranteed by constraining the Frobenius norm of the weight matrix W to be less than one". This would be true if the activation were contractive (a property which is later mentioned in two sentences), however it is not necessarily true in general. One needs the weight matrix to have a norm less than 1 AND to have a contractive activation.
- I am not sure what the value is in providing Figure 2 in the main paper. First, it is entirely qualitative. Second, it is only for the newly introduced DEQ model and not for the other models. In my opinion this could be left in the appendix, with a comparison with the other methods.

**Questions:**

- As far as I understand, one introduces a few DEQ layers (equation 5), which compute some candidate fixed point of the score function, which is then passed to the actual score function. Is that correct? How then does the DEQ (5) become close to the true score function? Are you just relying on the Fisher divergence loss?
- Is the following paper related at all? They compute fixed points of the score function using a DEQ, but for a rather specific model some kind of generalisation of PCA. Deep Equilibrium Models as Estimators for Continuous Latent Variables, AISTATS 2023
- It is mentioned that the Hessian is memory intensive. However, can't the HVP be computed without ever having to form the Hessian? This would then just be linear rather than quadratic in the number of parameters, without even having to approximate. I can understand time intensive, but not necessarily memory intensive. This is discussed somewhere in the third paragraph of the introduction and section 3, but could the authors please further clarify?

---

> ### Author Response · Authors · 2023-11-17
> **Response to Reviewer H3Py (1/2)**
>
> Thank you for your thoughtful review and valuable feedback, and for praising our paper as having good quality. In what follows, we provide point-by-point responses to your comments.
>
> ---
>
> **Q1: My biggest concern is related to my first Question in the box below. Why is this method actually principled? As far as I understand, a DEQ model is used as a kind of surrogate for the true score function, and then the fixed point of this surrogate is plugged into the score function and the Fisher Divergence is minimised. Does this result in the DEQ model being somehow close to the true score function, and in particular, the fixed point of the DEQ becomes close to the fixed point of the score function? I would appreciate it if the authors could spend some more time describing the method itself (i.e., I feel section 3.1 is too short).**
>
>
> **Reply:** As in traditional score-based density estimation models, the log density function is modeled using neural networks, with loss functions related to Fisher divergence terms like score matching (SM), denoising score matching (DSM), or sliced score matching (SSM).
>
> Our approach transforms the core layers of these models into DEQ layers. In this case, the fixed points act as an intermediate hidden representation. This modification significantly increases the model's depth, allowing for more complex representations while maintaining a low memory footprint. The score function, being the derivative of the log density function, is thus derived from the neural network's output, computed via autodifferentiation with Neumann approximation for efficiency.
>
> We have extended the discussion about the neural network modeling log density function and DEQ layer before Section 3.1 for clarity.
>
> ---
>
> **Q2: Proposition 2. It seems to require $K\rightarrow \infty$, however, the authors state they use $K=2$. There is no quantification of the error in finite settings. Furthermore, it is not clear how error in the derivatives translates to errors in the score, or errors in the Fisher divergence.**
>
>
> **Reply:** Thank you for your feedback on Proposition 2. In the revised version, specifically in Appendix C, we present a detailed empirical analysis aimed at evaluating the effectiveness of using $K =2$ for phantom gradients. This analysis includes a direct comparison between the phantom gradient approach (with $K=2$) and the exact gradient method for the DEQ block.
>
> We find that the score function remains largely consistent when employing phantom gradients with $K=2$, indicating that the approximation does not significantly deviate from the exact gradient in terms of both the values and trends. Moreover, we observed a strong correlation in the parameter update direction between the phantom gradient method and the exact gradient approach. This observation suggests that even with $K=2$, our method maintains a high degree of fidelity to the exact gradient.
>
> ---
>
> **Q3: The first sentence in the abstract doesn't seem to make sense, however I could be wrong. Double check with someone else. I would write "Score matching methods, which estimate probability densities without computing the normalization constant, are particularly useful in deep learning"**
>
>
> **Reply:** Thank you for your suggestion. We have updated our manuscript to improve the writing further.
>
> ---
>
> **Q4: I believe there is an incorrect statement in section 3.2. "The well-posedness of the fixed-point equation 5 can be guaranteed by constraining the Frobenius norm of the weight matrix W to be less than one". This would be true if the activation were contractive (a property which is later mentioned in two sentences), however it is not necessarily true in general. One needs the weight matrix to have a norm less than 1 AND to have a contractive activation.**
>
>
> **Reply:** That sentence is supposed to convey that the linear map inside the activation function is contractive. We have fixed the typo in presenting the linear map in the revision and clarified that contractive linear and non-expansive activation ensures the well-posedness of the fixed point equation 5.
>
> ---
>
> **Q5: I am not sure what the value is in providing Figure 2 in the main paper. First, it is entirely qualitative. Second, it is only for the newly introduced DEQ model and not for the other models. In my opinion this could be left in the appendix, with a comparison with the other methods.**
>
> **Reply:** Thank you for pointing this out. Indeed, these sampled images are only for visualization purposes. We have moved them from the main text into Appendix E. The FID scores presented in Table 1 present a better quantitative comparison.
>
> ---

---

> ### Author Response · Authors · 2023-11-17
> **Response to Reviewer H3Py (2/2)**
>
> **Q6: As far as I understand, one introduces a few DEQ layers (equation 5), which compute some candidate fixed point of the score function, which is then passed to the actual score function. Is that correct? How then does the DEQ (5) become close to the true score function? Are you just relying on the Fisher divergence loss?**
>
> **Reply:** As explained in the response to Q1. The original density estimation models output the log density function and use the Fisher divergence-related loss function for parameter optimization. Our DEQ integration replaces the core parts of the previous model with DEQ layers. The fixed point doesn’t necessarily have a meaning that is directly related to the score function but rather is just used as a hidden representation.
>
> ---
>
> **Q7: Is the following paper related at all? They compute fixed points of the score function using a DEQ, but for a rather specific model some kind of generalisation of PCA. Deep Equilibrium Models as Estimators for Continuous Latent Variables, AISTATS 2023**
>
>
> **Reply:** Thank you for pointing out this paper. This work has a different focus from ours. They focus on the maximum a-posteriori (MAP) estimation of the latent variables, observe that the MAP estimate satisfies a fixed-point equation, and then use the DEQ layer to model the latent variables. In contrast, we focus on the utilization of the DEQ layer in the case where the loss function contains derivatives, and DEQ is used to obtain memory-efficient gradient computation. We have added this paper to the related work section in the revised paper.
>
> ---
>
> **Q8: It is mentioned that the Hessian is memory intensive. However, can't the HVP be computed without ever having to form the Hessian? This would then just be linear rather than quadratic in the number of parameters, without even having to approximate. I can understand time intensive, but not necessarily memory intensive. This is discussed somewhere in the third paragraph of the introduction and section 3, but could the authors please further clarify?**
>
> **Reply:** The SM loss necessitates the calculation of the sum of squares of the diagonal elements of the Hessian matrix with respect to the input data. With HVP,  it can be computed with linear complexity in data dimension. Not only will it be time-consuming but also memory-intensive. The memory bottleneck comes from the size of retained computational graphs for computing  Hessian-related terms. The need for retention stems from the presence of Hessian-related terms in the loss function. Consequently, retaining these computational graphs becomes necessary for updating the model's weights during the backpropagation.
>
> ---
>
> We have updated our submission based on the reviewer's feedback, with the revision highlighted in blue. We are happy to address further questions on our paper. Thank you for considering our rebuttal.

---

### Author Response · Authors · 2023-11-17
**General Response**

Dear Reviewers and AC,

We sincerely thank you for your thorough and insightful reviews of our paper. Your constructive feedback has been invaluable in identifying areas for improvement and clarification. We are grateful for the opportunity to address your concerns and clarify aspects of our work.

Firstly, we would like to express our gratitude for acknowledging the potential and quality of our paper. We appreciate your recognition of the novelty in combining Deep Equilibrium Models (DEQs) with score-matching techniques, an approach not explored in previous literature. We believe that our work contributes to the field by addressing the crucial challenges in score-matching-based density estimation and generative models, particularly in managing higher-order derivatives efficiently during backpropagation.

---

**Addressing common comments**

- Regarding the novelty of the integration of DEQ into score matching-based estimation. As far as we are aware, our work is the first one that studies DEQ in the setting of score matching that requires the computation of higher derivatives.

- Regarding concerns about the error implications of using the phantom gradient method with $K=2$, we have included additional numerical evidence in Appendix C. This evidence supports our usage, demonstrating that $K=2$ is a simple and effective choice for phantom gradients in our context.

- Additionally, we acknowledge that many DEQ techniques, such as the advanced DEQ model MonDEQ, advanced fixed-point solvers and their accelerations, and refined versions of phantom gradient, can potentially be applied to our proposed DEQ-based score-matching framework. We didn't include these techniques in our paper as we want to focus on the core idea of integrating DEQ into score matching. We hope that our work will inspire future research in this direction.

---

**Summary of the major revision**

Incorporating the comments and suggestions from all reviewers, we have fixed typos and reformated the paper. Moreover, we have made the following major changes in the revision:

- We have clarified the regularity assumption in our propositions; in particular, the employed activation functions are smooth when SM loss is needed.

- We provide an empirical analysis of phantom gradients in Appendix C.

- We extend the experiment-related details in Appendix D.

- We add convergence-related hyperparameters and results for solving the fixed point in Appendix F.

---
We have responded to all the reviewers' comments in detail in the following sections. We hope that our responses will address your concerns and that you will find our revised manuscript merits publication in ICLR 2024. Thank you for considering our rebuttal.

---

### Public Comment · ~Tianxiang_Gao2 · 2023-11-19
**Related Contributions to Your Research**

Dear Authors,

I wanted to draw your attention to a couple of significant research contributions that seem to relate closely to the results presented in your paper. I believe these works are pertinent to your findings but have been omitted from your references.


In our work [1], we utilized random matrix theory to demonstrate, theoretically, that using a fixed scalar can guarantee DEQs' well-posedness throughout training, not just at initialization. This approach enabled us to establish global convergence results for gradient descent in training deep equilibrium models.

Furthermore, as DEQs utilize shared parameters, their expressive capacity is predominantly governed by their width, that is, the number of parameters or neurons in each layer. In [2], our research demonstrates that as the width tends towards infinity, DEQs exhibit equivalence to a Gaussian process—a relationship commonly known as the NNGP correspondence. Additionally, we showcase that the corresponding covariance function or NNGP kernel is strictly positive definite. As highlighted in previous studies, a strictly positive definite NNGP kernel plays a significant role in ensuring the convergence of gradient-based methods for DEQs and estimating generalization on previously unseen data. These insights play a crucial role in understanding and ensuring the stability and robustness of DEQs during training and beyond.

I believe these works align closely with the themes and outcomes discussed in your paper and would be valuable additions to your related literature.

Best regards,

Tianxiang

[1] Gao, Tianxiang, et al. "A Global Convergence Theory for Deep ReLU Implicit Networks via Over-parameterization." ICLR 2022.

[2] Gao, Tianxiang, et al. "Wide Neural Networks as Gaussian Processes: Lessons from Deep Equilibrium Models." NeurIPS 2023.

---

> ### Author Response · Authors · 2023-11-19
> **Thank you for pointing out these papers to us**
>
> Dear Tianxiang,
>
> Thank you for bringing these two papers to our attention, and we will carefully read and digest them first.
>
>
> Regards,
>
> Authors

---

> ### Comment · Reviewer_YwTh · 2023-11-21
>
> Hi Tianxiang,
>
> I think your paper [1] also has an error in the proof of Lemma 2.3 (the same error that was made in this submission before revision). You apply implicit differentiation to a nonsmooth function (ReLU) and this is not correct without a nonsmooth implicit function theorem; the solution mapping is not always differentiable. The typical implicit function theorem requires that the function you are implicitly differentiating is **continuously differentiable** in a neighborhood of the equilibrium (thus excluding ReLU without further justification). Compare for instance with "Monotone operator equilibrium networks" by Winston and Kolter or "Nonsmooth Implicit Differentiation for Machine Learning and Optimization" by Bolte, Le, Pauwels, and Silveti-Falls, both of which show how to use a nonsmooth implicit differentiation theorem to correctly apply implicit differentiation to DEQs with nonsmooth activations.

---

### Meta-Review · Area_Chair_6wLD · 2023-12-05

**Metareview:**

This paper proposes to use Deep Equilibrium Models (DEQs) in the context of score-matching generative models, to improve their memory and time efficiency. Through theoretical results and a thorough experimental framework, it demonstrates the advantage of DEQ-based score-matching models over non-DEQ ones for a fixed computational and memory budget.

There was consensus among the reviewers that the paper is clear, well-written, and that the method is sound and appropriately evaluated. The reviewers had many techincal questions, most of which were resolved in the discussion with the authors. One of those involved an error in the proof of the main result, which required relaxing the generality of the setting (by assuming the activation function is twice continuously differentiable). After this modification, the reviewer that pointed out this flaw seems to have been satisfied that the results were reasonably salvaged. After the rebuttal phase, it appears that the lingering weakness are more fundamental and related to the novelty of originalty of the contribution. These, however, are outweighed by the strengths of the paper, hence why I recommend acceptance.

**Justification For Why Not Higher Score:**

The strength and novelty of the contributions do not, in my opinion, amount to a spotlight or oral.

**Justification For Why Not Lower Score:**

The paper is sound, throught, and interesting enough to warrant acceptance.

---

### Decision · Program_Chairs · 2024-01-16

Accept (poster)